

# High resolution continuous flow analysis impurity data from the Mount Brown South ice core, East Antarctica

Margaret Harlan[1,2,3], Helle Astrid Kjær[2], Aylin de Campo[2,4], Anders Svensson[2], Thomas Blunier[2], Vasileios Gkinis[2], Sarah Jackson[5,6], Christopher Plummer[7,1], and Tessa Vance[1]

[1]Australian Antarctic Program Partnership, Institute for Marine and Antarctic Studies, University of Tasmania, nipaluna/Hobart, Tasmania, Australia.
[2]Physics of Ice, Climate and Earth, Niels Bohr Institute, University of Copenhagen, Copenhagen, Denmark.
[3]Institute for Marine and Antarctic Studies, University of Tasmania, Battery Point 7004, Australia
[4]Victoria University of Wellington, Antarctic Research Centre, Wellington, New Zealand
[5]Climate and Environmental Physics, Physics Institute, University of Bern, Sidlerstrasse 5, Bern, Switzerland
[6]Oeschger Centre for Climate Change Research, University of Bern, Hochschulstrasse 4, Bern, Switzerland
[7]Australian Antarctic Division, Department of Climate Change, Energy, the Environment and Water, Kingston, Tasmania, Australia

**Correspondence:** Margaret Harlan (margaret.harlan@utas.edu.au)

**Abstract.** The Mount Brown South ice core (MBS 69.111 °S 86.312 °E) is a new, high resolution ice core drilled in coastal East Antarctica. With mean annual accumulation estimated to be 20-30 cm ice equivalent accumulation throughout the length of the core (∼295 m), MBS represents a high resolution archive of ice core data spanning 1137 years (873 - 2008 CE), from an area previously underrepresented by high resolution ice core data.

5 Here, we present a high-resolution dataset of chemistry and impurities obtained via continuous flow analysis (CFA). The dataset consists of meltwater electrolytic conductivity, sodium ($Na^+$), ammonium ($NH_4^+$), hydrogen peroxide ($H_2O_2$), and insoluble microparticle measurements. The data are presented in three datasets: as a 1 mm depth resolution record, 3 cm averaged record, and decadal average record. The 1 mm record represents an oversampling of the true resolution, as due to smoothing effects the actual resolution is closer to 3 cm for some species. Therefore, the 3 cm resolution dataset is considered

10 to be the minimum true resolution given the system setup. We also describe the current Copenhagen CFA system, and provide a detailed assessment of data quality, precision, and functional resolution.

The 1 mm averaged, 3 cm averaged, and MBS2023 decadal averaged datasets are available at the Australian Antarctic Data Center: http://dx.doi.org/doi:10.26179/9tke-0s16 (Harlan et al., 2024).

## 1 Introduction

15 Ice cores represent some of the best proxies available for reconstructing past atmospheric composition, in terms of atmospheric gases as well as atmospheric aerosols. While atmospheric gases are contained and preserved in air bubbles in the ice, aerosols can be preserved as soluble and/or insoluble compounds deposited by dry deposition onto the snow surface or with the snow as wet deposition, and incorporated into the ice matrix as the snow is compressed into ice, (Legrand and Mayewski, 1997). The



chemical compounds from such aerosols can be measured in the ice itself, either by the chemical composition of the ice (ionic species), or by measuring the insoluble particulate material in the ice.

Many of the aerosol impurities contained in ice cores reflect seasonal variability of aerosol species. However, reconstructing seasonal cycles requires sampling resolution high enough for the seasonal signal to be resolved. This can present measurement challenges, as discrete measurements of trace chemistry of ice samples can require intensive manual decontamination of sampling, which is time consuming and can be difficult to achieve under cold laboratory conditions. Decontamination is especially labor intensive when centimeter-scale measurements are required for ice cores that range from hundreds to thousands of meters in length. Continuous flow analysis (CFA) chemistry measurements, pioneered by Sigg et al. (1994), allows entire lengths of ice core to be sampled continuously in high resolution with very minimal sample decontamination required (Bigler et al., 2011; Kaufmann et al., 2008; Erhardt et al., 2022).

The new Mount Brown South ice core (MBS), one of only two millennial length ice cores in East Antarctica (together with the Law Dome ice core), represents an important new paleoclimate record from an under-represented region in Antarctica (Vance et al., 2016, 2024a). Data from the upper (satellite-era) sections of the MBS ice cores has been used to investigate an East Antarctic proxy for El Niño variability (Crockart et al., 2021), as well as to understand precipitation climatology effects on the water-isotope record for the coastal East Antarctic site (Jackson et al., 2023). The full length of the MBS core has been dated and the chronology (MBS2023, used here) is described in Vance et al. (2024a).

Here we present the high-resolution, CFA record of chemistry and insoluble impurity measurements for the full length of the MBS ice core. The data set is presented alongside a description of the analytical system and methodology, as well as considerations about data quality and uncertainties in both the concentration and the depth/temporal resolution of the record.

## 1.1 Continuous Flow Analysis

CFA is particularly well suited for measuring trace chemical species and impurities at high resolution (millimeter to centimeter scale) for long ice cores (Bigler et al., 2011). This allows for the production of seasonal-scale records of aerosol species spanning thousands of years at sites with sufficiently high accumulation. CFA is advantageous compared to discrete methods such as ion chromatography, as it provides the ability to analyze ice core samples at speeds of up to four centimeters per minute and with minimal time required for decontamination and sample changeover.

The CFA process involves melting a vertical section of an ice core (typically from top to bottom) by placing it on a purpose-built heated plate (melthead) connected to a melt water extraction system. The melthead is designed to prevent contamination, such that the external portion of the core section is diverted to waste and thus manual decontamination is only required along the horizontal surfaces at core breaks. The inner decontaminated melt water is debubbled and subsequently diverted through a system of measurement instrumentation using a series of peristaltic pumps. Measurements are taken continuously (data recorded each second). The detectors used are chosen for their short response time and ability to detect the low impurity concentrations observed in polar ice (Sigg et al., 1994).

The modular nature of CFA allows it to be simplified for deployment to the field for in-situ measurements (Kjær et al., 2021). It may also be expanded for additional gas (Stowasser et al., 2012; Rhodes et al., 2013) and isotope (Gkinis et al., 2011;

Jones et al., 2017) measurements in addition to chemistry and impurities (Bigler et al., 2011). The Copenhagen CFA setup used in the 2018 and 2019 MBS campaigns both build on the CFA setup described in Bigler et al. (2011) and Kjær et al. (2022).

Both campaigns used the same system components, however, the entire CFA system was relocated to a new building between campaigns. The move resulted in some differences between the setups, which are detailed below.

The chemistry and impurity measurements from both campaigns include insoluble microparticles (dust > 1 $\mu$m), electrolytic conductivity, calcium ($Ca^{2+}$), sodium ($Na^+$), ammonium ($NH_4^+$), hydrogen peroxide ($H_2O_2$), and meltwater acidity. Additionally, sulfate ($SO_4^{2-}$) was analysed in 2018. The acidity, sulfate, and calcium records, however, are not presented here, due to

independent problems with the measurement systems for those species.

## 1.2    Ice core site

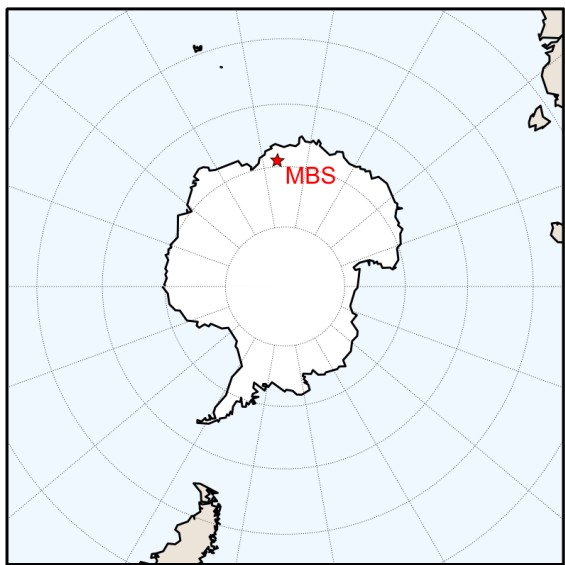

**Figure 1.** Map showing the location of the Mount Brown South site.

Four ice cores were drilled during the 2017-2018 austral summer field season at the coastal East Antarctic Mount Brown South site (MBS1718-Main, -Alpha, -Bravo, and -Charlie). The MBS-Main drill site lies at 69.111°S, 86.312°E, 2084 meters ASL, adjacent to the boundary between Princess Elizabeth and Wilhelm II Land, approximately 62 km south of the Mount

Brown nunatak, and located 12.6 km WSW of a short surface core drilled in December 1998 (known as "MBS99," Smith et al. (2002); Foster et al. (2006)). While the four cores drilled during the 2017-2018 season are named MBS1718, for simplicity, we refer to them here as simply MBS. The MBS site was selected based on six criteria as described in Vance et al. (2016) in order to (1) preserve a millennial-scale record in the 300 meter archive; (2) undergo minimum snowmelt in summer; (3) preserve sub-annual resolution (minimum accumulation of 25 cm/yr ice equivalent); (4) experience minimal surface reworking;

(5) have undergone minimal ice advection at 300 m depth; and (6) add novel information to the existing network of East Antarctic ice cores. Average accumulation at the site is found to be 30 cm ice equivalent per year over the satellite era (1978 to 2017, based on annual layer depth counting of the shallow cores) (Crockart et al., 2021) and the site is characterized by wet deposition (Crockart et al., 2021; Vance et al., 2024a). Mean annual surface air temperature at the MBS site is -29.7°C (based on the Modèle Atmosphérique Régional (MAR) regional climate model, Agosta et al. (2019)), and the site is characterized by

predominantly easterly prevailing winds (Vance et al., 2024a). Drill site characteristics for MBS are presented in Vance et al. (2024a) and detailed climatology of the site is well described by Jackson et al. (2023) and Crockart et al. (2021).

    Vance et al. (2024a) presents the MBS2023 chronology, a combined depth-age scale for MBS1718-Main and MBS1718-Charlie. The 295 meters of the main core cover 1137 years (873-2008 CE), while the three co-located surface cores (Alpha, Bravo, and Charlie) cover approximately the satellite era through to drilling year (2017-2018).

**2   Sample description**

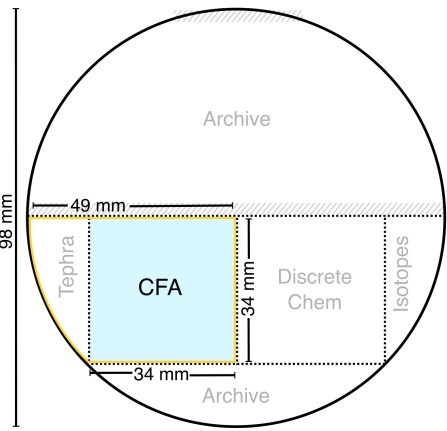

**Figure 2.** Schematic diagram showing the cross-section of the MBS main core, with sample sections shown. Shaded areas indicate cuts made with a bandsaw. Dashed lines indicate surfaces planed for intermediate layer core scanning. Section shipped to Copenhagen shown outlined in yellow, with CFA stick shaded in blue.

    The core was drilled as part of a joint Australian-Danish collaboration using the Danish Hans Tausen ice core drill (9.8 cm core diameter). The ice was transported from the drill site to Hobart, where it was described (in terms of core quality, including breaks, cracks, and other damage to the core), logged, and imaged in the -18°C freezer laboratories at the Institute for Marine and Antarctic Studies. There, it was sectioned lengthwise for analyses. An interior piece with 34 x 34 mm cross section area

along the entire length of core was designated for CFA chemistry (Fig. 2). See Vance et al. (2024a) for a full description of the ice processing.

    The CFA section, still attached to the wedge-shaped outer edge piece designated for tephra sampling, was transported to Copenhagen, where the samples were prepared for CFA by removing said outer wedge at freezer facilities at the Physics for





Ice, Climate and Earth (PICE) at the Niels Bohr Institute (NBI). To minimize potential contamination, the horizontal ends of

each sample piece, including any breaks occurring within a sample, were carefully cleaned by scraping with a ceramic blade

immediately prior to being placed in frames for melting on the Copenhagen CFA.

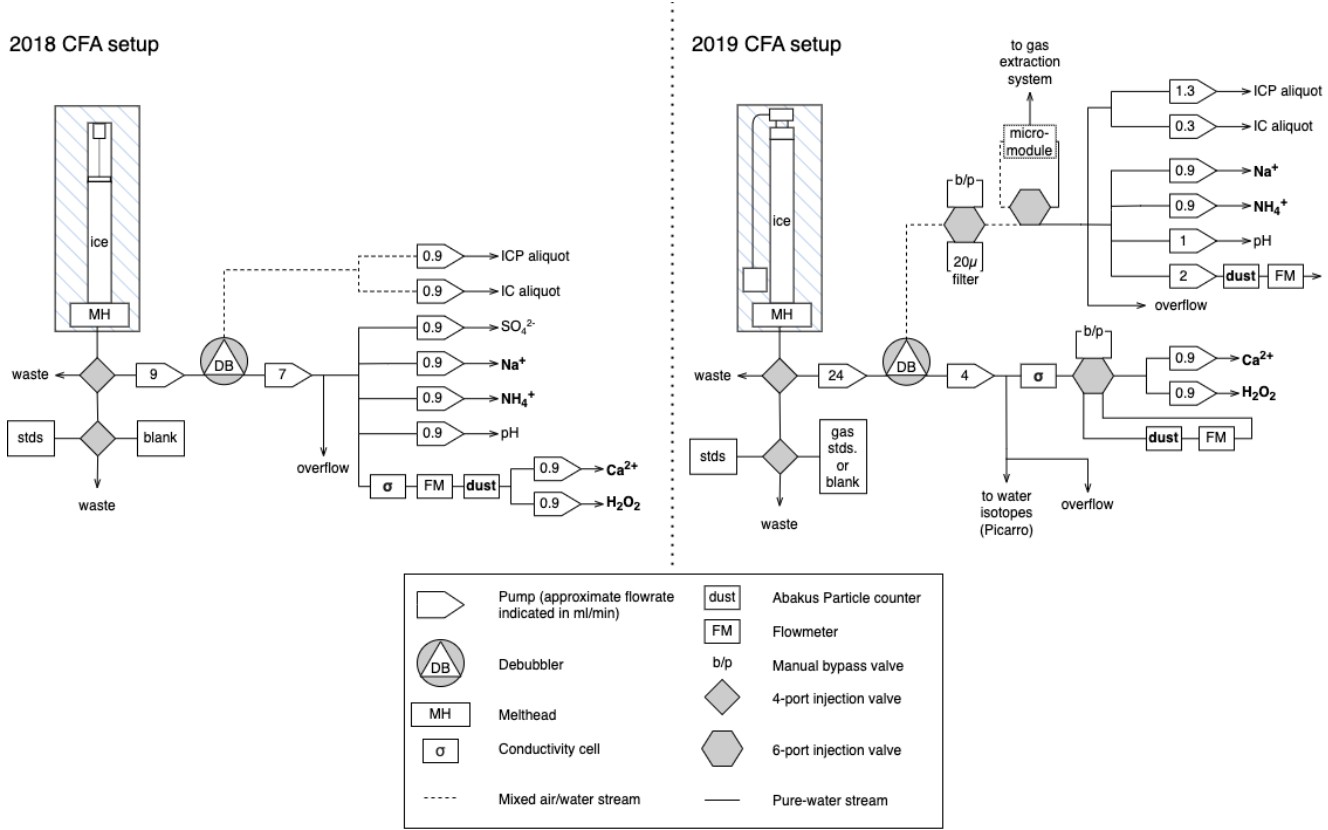

**Figure 3.** Schematic flow diagram of the CFA systems used for the 2018 and 2019 measurement campaigns at the Physics of Ice, Climate, and Earth in Copenhagen.

## 3    MBS continuous flow analysis

The basic setup of the Copenhagen CFA system is well described in Bigler et al. (2011) and Kjær et al. (2022), and is based on

previous CFA systems such as Kaufmann et al. (2008) and Röthlisberger et al. (2000). The CFA measurements took place over

two sampling campaigns. The dry-drilled section (∼4-94 m) of the main core, and the Charlie core (25.86 m) were analyzed in

2018. The remaining (wet-drilled) section of the main core (∼95-295 m) was analyzed in 2019 after the CFA laboratory was

relocated to another building. Flow diagrams for the two systems used for the MBS CFA measurements can be seen in Fig. 3.

Details of adaptations to the system made between the two campaigns are described below.



The CFA base functionalities are very similar between the two analytical campaigns: Sticks of firn/ice are placed into
polycarbonate frames for mounting above the melter system consisting of the heated melthead (Bigler et al. (2011)) and above
it a short ice alignment guide (~20 cm), all held inside a freestanding freezer unit maintained at approximately -20°C. The
frames and alignment guide allow for each new stick of ice to be added to the system before the previous stick is fully melted.
This allows for continuous melting of multiple (typically 5-10) meters of ice during each run similar to the CFA procedure
described in Erhardt et al. (2023) and Kjær et al. (2022).

As the ice is melted, meltwater from the clean interior of the sample is pumped to the analytical systems using peristaltic
pumps (ISMATEC), while water from the potentially contaminated outer ice is diverted to waste. Both setups employ the use
of an enclosed debubbler consisting of a flat triangular cell to remove air bubbles from the sample stream (Bigler et al., 2011).
Both systems utilize a commercially available conductivity meter to measure meltwater electrolytic conductivity, as well as
an Abakus laser particle counter (LDS23/25bs; Klotz GmbH, Germany) coupled with a flowmeter to determine the number
of insoluble particles larger than 1 μm per ml of sample volume (Simonsen et al., 2018). Simple fluorescence or absorption
spectroscopic methods are used to determine $NH_4^+$, $Ca^{2+}$, $H_2O_2$, $Na^+$, $SO_4^{2-}$, and meltwater acidity, using instrumentation and
methods specifically designed for CFA (Kaufmann et al., 2008; Sigg et al., 1994; Röthlisberger et al., 2000; Erhardt et al., 2022;
Kjær et al., 2022; Kjær et al., 2016). Reagents and buffers for analysis were prepared weekly or bi-weekly as required with
respect to their rate of deterioration. Additionally, during both campaigns, aliquots were collected in vials for later analysis
using ion chromatography (IC) and inductively coupled plasma mass spectrometry (ICP-MS), the results of which are not
included in this dataset and will be released in a future publication.

### 3.0.1 Specifics for the 2018 CFA setup

The 2018 CFA measurements took place at the Center for Ice and Climate at the University of Copenhagen in the laboratory
described also by Simonsen et al. (2019) (which at that point was located at Juliane Maries Vej Copenhagen) during October-
November 2018.

An optical depth registration system (Waycon LLD-150-R5232-50H) was used for registration of the melt speed during the
2018 campaign, in a setup similar to that used in (Dallmayr et al., 2016) While the laser distance meter allows for precise melt
speed and sample length measurements ($\pm$ 2mm accuracy, and 0.5 mm resolution), it can be temperamental as vibrations from
ambient activity (heavy footfalls or accidental knocking of the freezer enclosure) can disrupt measurements. Additionally, the
ice is never a perfect fit to the frame and as it melts it changes from leaning on one side of the frame to the next causing some
additional noise seen in the very precise laser measurement. Additionally, the space required for the laser system limits the
length of the frames used in the freezer, thus limiting the sample stick length to 50 cm. For the campaign in 2018, 10 sample
sticks (5 m) were melted continuously per run, book-ended by standards runs (see section 4.3 and table A1).

A sulfate measurement system following the method used for NorthGRIP (Kaufmann et al., 2008; Erhardt et al., 2022) was
trialed in the 2018 system, but due to a high limit of detection of the system, sulfate measurements were unsuccessful, and thus
are not presented here. Discrete sulfate measurements of the full MBS record via ion chromatography can be found in Vance
et al. (2024a).



### 3.0.2 Specifics for the 2019 CFA setup

In October-November 2019 the Physics of Ice, Climate, and Earth at the University of Copenhagen moved to a new address
at Tagensvej 16. Due to the relocation the 2018 system was dismantled and rebuilt in the new location. During reassembly changes were made to the system, described herein.

    A cable-driven rotary encoder (draw wire position transducer, SX80, WayCon) was implemented for depth registration in 2019 similar to that described in Bigler et al. (2011). This encoder setup freed up more space in the freezer unit, allowing for melting of meter long sample sticks and thus the data from the 2019 campaign has fewer sample breaks per run as 5-10 samples
(5-10 m) were melted continuously per run. Further, all chemistry lines were optimized in length to minimize dispersion and thus smoothing of the final record.

    As part of the 2019 rebuild, a gas extraction system (3M, Liqui-Cel MM-0.75×1 Series) coupled to the mixed air-water stream from the top of the debubbler unit was added to the CFA. While the gas extraction system was not used for data collection during the MBS melting campaign, it was sporadically implemented for testing and refinement throughout the 2019
measurements. Further detail on the gas extraction system follows below.

## 4 Data processing

### 4.1 Depth scale

Precise depth information is crucial for ice core data. The position derived melt-rate data collected during CFA campaigns is used to reconstruct the total length of ice melted, to which the recorded length of ice removed during preparation must be
re-introduced at the appropriate depths. To account for the time when the encoder is removed during the frame changes, for both the laser and draw-wire encoder, the position data is used to identify the times just before and after each frame change period, and the average meltrate over the previous period of uninterrupted melting is used to reconstruct a continuous melt-rate.

    The information recorded during sample preparation is used (including ice removed for decontamination at sample ends and breaks and poor-quality ice not analyzed), together with the position of each break as logged during melting, to create
appropriately sized gaps in the depth scale corresponding to the missing ice. The depth scale is finalized by assigning the top of each CFA data run to the recorded top depth of the corresponding bag from the agreed upon field depth measurements. Manual decontamination of the cores can introduce slight (sub-millimeter) discrepancies due to mis-reading lengths on the standard ruler used in preparation as well as slight misjudgments and biases in identification of break positions during melting. These unavoidable inaccuracies are estimated to be on the scale of one or two millimeters at most.

In rare cases, measured lengths of the same stick varied from one measurement to another (for example, when measured in Hobart prior to shipping to Copenhagen). This was found to be due to slightly slanted cuts where one core meets the next, with one measurement taken from the long side and another taken from the short side. These discrepancies were found to amount to less than 0.01% of the length of the record. Careful consultation of comprehensive line scan images of the core sections were





used to correct for these errors and determine an agreed upon core length across all measurements. This procedure is described
in detail in Vance et al. (2024a).

## 4.2 MBS Chronology

An age scale has been developed for the MBS ice cores, described thoroughly in Vance et al. (2024a). The ice core dating and
layer counting was based on a combination of stable water isotopes (Gkinis et al., 2024) and discrete chemistry measurements,
primarily the ratio of sulfate to chloride, and there was found to be significant variability in annual layer thickness throughout
the length of the core. The chronology was determined by two independent layer counting efforts, one relying on volcanic
matching, the other primarily relying on variable chemistry species. These two efforts were followed by careful consideration
and joint determination of uncertain years (Vance et al., 2024a). We encourage direct users of the MBS CFA dataset to use the
MBS2023 chronology (Vance et al., 2024b), or any subsequent updates thereafter.

## 4.3 Calibrations

The absorption and fluorescence spectrometric methods used for chemistry concentration measurements require calibration
to convert from instrument signal voltage/light intensity to concentration. In order to properly calibrate these instruments,
standard solutions with predetermined chemical concentrations are passed through the system before and after each sample
run. A multi-element standard solution is used for a three step calibration of the $Ca^{2+}$, $Na^+$, and $NH_4^+$ measurements (Merck
Certipur®IC Multi-element Standard VII). A single component standard solutions is used for a two step calibrations of $H_2O_2$
(Sigma-Aldrich Hydrogen Peroxide Solution 30 wt % in $H_2O$). The multi-element standard solutions are prepared prior each
run, and the peroxide standards are prepared prior to each run from a first dilution prepared at the start of each measurement
day. All standards are prepared using ultra-pure deionized water (Merck RiOs™16 MilliQ). Standard recipes and resulting
concentrations are presented in Table A1.

Calibrations are calculated from the standards run at the beginning and end of each measurement run. Each standard solu-
tion is passed through the system in series and the resulting signal voltage was recorded both manually and by the LabVIEW
software used to operate the system. For the fluorescence method ($Ca^{2+}$, $NH_4^+$, and $H_2O_2$ measurements), the standard calibra-
tion is based on the linear relationship between concentration and fluorescence signal voltage. Sodium is a pseudo-absorption
method, and calibration is based on a curve fit to the standard signal response (Kaufmann et al., 2008). Calibrations are
computed using a semi-automated script written in MATLAB. The script isolates the signal voltage at each standard input
concentration and calculates the appropriate calibration coefficients. The standards are well fitted, with the average r-square
value for the calibration curves for each analyte being greater than 0.99.

## 4.4 Signal delay time

Due to differing distances of each of the measurement instruments from the melter, there is a time delay between when the ice
passes the melter and when the measurement takes place.

The time delay between the ice on melter and when the measurement is recorded down stream is calculated in two parts. Firstly, there is the elapsed time from when the sample parcel passes the melter to when it is divided into disparate melt streams for each analytical unit. The bulk melt delay time is measured during sample melting as the elapsed time from when the first sample ice passes the melter (recorded as observed by lab operator) and the initial peak response observed in the conductivity measurements (15 seconds for both systems). Conductivity is used as it is located closest to the melter (by tubing

distance/mixing volume as well as time), and thus the first signal to respond.

Secondly, there is a delay time individual to each species measured due to the internal dynamics of each analytical melt stream. These response delay calculations are computed based on each instrument response time during the standard run. This additional delay time is measured as the time elapsed between when the signal increase is seen in the conductivity measurements and target species. We determined that additional delay time as time at which the derivative of the response

curve reaches a maximum (approximating the midpoint of the sample response rise). Delay time varies for each species: $79 \pm 3.6$ seconds for $Ca^{2+}$, $73 \pm 5.9$ seconds for $NH_4^+$, $48 \pm 5.3$ seconds for $H_2O_2$, and $68 \pm 4.0$ seconds for $Na^+$.

### 4.5    Calcium data

Expected concentrations of calcium at the MBS site are very low (median chemistry concentration from the discrete chemistry measurements is 0.142 ppb). Due to the nature of the system, and the limits of the ultrapure water system used to produce

the standards, reagents, and baseline values for measurements produced here, this is significantly below the limit of detection we are reliably able to achieve using the Copenhagen CFA system (see Table 2). As the CFA $Ca^{2+}$ data for MBS does not show any discernible seasonal cycles throughout the record, and the calibrated $Ca^{2+}$ concentration values measured close to or below the LOD of our instrumentation, we consider these values too low to report.

## 5    Data resolution and smoothing

### 5.0.1    Sampling resolution

Data is registered at one second time steps, and thus the sampling resolution varies with meltrate. The meltrate of the CFA system is selected to optimize resolution, however with the introduction of the gas extraction system in 2019, the meltrate was calibrated to produce optimal gas volumes for simultaneous measurement. The 2018 system setup (5 - 95 m depth) operated with a target meltrate of 3.5 cm min$^{-1}$ (median actual metlrate 3.45 cm min$^{-1}$). The 2019 campaign (96-295 m depth) had a

target meltrate of 4 cm min$^{-1}$ (median actual meltrate 4.08 cm min$^{-1}$. These meltrates produce a direct sampling resolution (independent of smoothing due to response time) of 0.58 and 0.68 mm, respectively.

### 5.0.2    Signal dispersion smoothing

The internal dynamics of the system (mixing volumes and flow conditions within tubing) lead to smoothing due to signal dispersion, as described by Breton et al. (2012). This gives a smoothing effect, wherein a discrete parcel of sample is measured



**Table 1.** Response time as 10%-90% rise time ($t_{10-90}$) and $e$-folding time ($\tau_e$) and resultant effective smoothing length based on meltrate (resolution). Values shown for each of the two system setups from 2018 and 2019 (and the depths measured during each campaign).

|  | **2018** (5-95 m depth) (median meltrate = 3.45 cm min$^{-1}$) | | | **2019** (95-295 m depth) (median meltrate = 4.08 cm min$^{-1}$) | | |
|---|---|---|---|---|---|---|
|  | $t_{10-90}$ (s) | $\tau_e$ | resolution (cm) | $t_{10-90}$ (s) | $\tau_e$ | resolution (cm) |
| Conductivity | 17.5 | 8.0 | 1.01 | 9.8 | 4.4 | 0.67 |
| NH$_4^+$ | 41.7 | 19.0 | 2.39 | 35.3 | 16.1 | 2.40 |
| H$_2$O$_2$ | 52.9 | 24.1 | 3.04 | 41.0 | 18.6 | 2.79 |
| Na$^+$ | 41.6 | 18.9 | 2.39 | 33.3 | 15.1 | 2.24 |

in the CFA System as a dispersed signal spread across a short time period (on the order of a few seconds) spanning the expected signal response time for that parcel (Breton et al., 2012). This dispersion time is calculated as as an $e$-folding time calculated based on the 10 - 90% rise time in the signal response during the standards runs. Using the meltrate, it is possible to use this dispersion signal time to calculate the effective smoothing length (and thus a minimum realistic resolution) for each species (Table 1). Although the meltrate used in 2019 was faster than in 2018 (4.08 and 3.45 cm min$^{-1}$ respectively), the shorter response times (likely due to shorter tubing lengths used in the rebuilt system) resulted in a higher resolution in 2019.

It is worth noting that, as described by Breton et al. (2012), the signal peak under realized (dispersed) flow conditions occurs slightly before the expected signal response under idealized "plug flow" conditions. This effect is linked to the specific dynamics of each CFA system, and similar to Erhardt et al. (2022), we do not quantify this slight offset for the Copenhagen system, but note that it could lead to a slight systematic offset (on the order of a few millimeters) biased towards deeper sample depths.

## 6 Uncertainties and data quality

### 6.1 Gas extraction system interference

During the 2019 CFA melting campaign, the introduction of a new gas extraction system was trialed. This system comprised a gas extraction line originating from the upper outlet of the debubbler, coupled to a micro-module which extracts dry air from the sample stream for gas analysis. These system trials influenced the internal pressure balance of the chemistry system tubing. This had a significant impact on the meltwater acidity (pH) measurements (Kjær et al., 2016), which suffered from back pressure changes influencing the sensitive dye to water ratios, making the pH record unusable. The gas extraction system also impacted the insoluble microparticle data collected by the ABAKUS laser particle counter.

While the specifics of the gas extraction system are beyond the scope of this data description, there is a visible signature in the microparticle record that only occurs when the gas extraction system was on-line with the chemistry CFA. The signature can be seen as a significantly elevated baseline in the microparticle record, as well as an alteration in the amplitude of the





**Table 2.** Sample data range and LOD for each analyte.

|  | **2018** (5-95 m depth) | | | **2019** (95-295 m depth) | | |
|---|---|---|---|---|---|---|
|  | Range (5th & 95th percentile) | Median | LOD | Range (5th & 95th percentile) | Median | LOD |
| Conductivity ($\mu$S cm$^{-1}$) | 0.74 - 1.55 | 1.03 | 0.01 | 0.57 - 1.38 | 0.85 | 0.05 |
| Dust (# ml$^{-1}$)* | 97.32 - 707.20 | 281.99 | 25.00 | - | - | - |
| Ca$^{2+}$ (ppb)* | - | - | 0.82 | - | - | 0.87 |
| NH$_4^+$ (ppb) | 0.07 - 1.03 | 0.32 | 0.03 | 0.09 - 0.87 | 0.34 | 0.12 |
| H$_2$O$_2$ (ppb) | 6.84 - 69.92 | 26.82 | 0.65 | 5.40 - 49.01 | 21.12 | 1.28 |
| Na$^+$ (ppb) | 2.39 - 38.89 | 13.54 | 4.26 | 3.01 - 36.41 | 14.11 | 3.94 |

* Dashes indicate data not included in the published dataset.

dust signal. These changes correspond with the recorded valve switches of the gas system. Due to the nature of the system interference, we are not able to reliably correct for this interference. We therefore only present the insoluble microparticle data for the dry drilled section (5-95 m), before the gas system testing began.

## 6.2 Analytical precision

The limit of detection (LOD) varies for each detection channel. LOD is calculated as 3 times the standard deviation of the blank signal measured on ultrapure (MilliQ) water during the standard runs (Röthlisberger et al., 2000; Gfeller et al., 2014). Sample concentration ranges and limit of detection (LOD) for each of the analytical detection channels is presented in Table 2. Frequency histograms for each measured species are presented in Fig. 4

## 6.3 Conductivity peak matching

To verify data quality and depth scale accuracy, we investigated all conductivity peaks that fall above $3\sigma$ from a 90-cm (approximately 3 year) moving mean of the conductivity record. Using this method, we are able to identify many of the volcanoes reported in Vance et al. (2024a) to within one year age-at-depth uncertainty. Volcanoes identified in this method include Pinatubo (1991), Agung (1963), Tambora (1815), Mount Mélébingóy/Parker Peak (1640), Ruiz (1595), and Samalas (1257), in addition to a number of the unknown peaks identified in MBS (Table A2). Additional peaks above $3\sigma$ exist, however have not been matched to volcanoes identified in Vance et al. (2024a).

## 6.4 Standards preparation

Measurement uncertainty for the wet chemistry analyses is driven primarily by uncertainty in standards preparation, due to instrument uncertainty of the microliter pipette (20-200 μl Socorex Acura manual®) and bottle-top dispenser (Dispensette®III




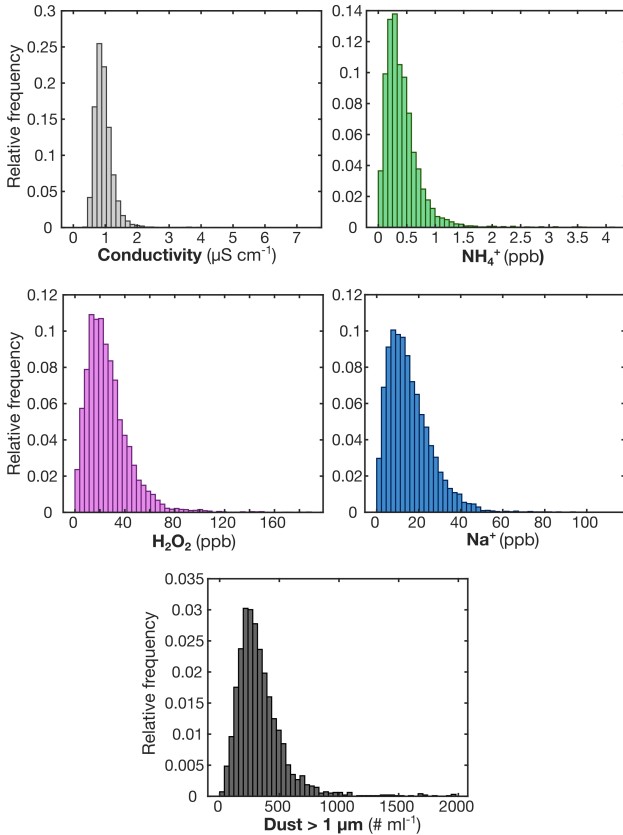

**Figure 4.** Relative frequency histograms demonstrating the variability of the impurity data (conductivity ($\mu$S cm$^{-1}$), peroxide (H$_2$O$_2$, ppb), sodium (Na$^+$, ppb), ammonium (NH$_4^+$, ppb), and insoluble microparticles (dust, particles per ml >1$\mu$m)). Histograms computed based on 3 cm averaged resolution dataset.

5-50 ml, Brand GmbH). This uncertainty is discussed in Gfeller et al. (2014), and as standards used here were prepared in a similar manner, the uncertainty estimate is less than 10% (Gfeller et al., 2014; Erhardt et al., 2023).

### 6.5 Contamination and data cleaning

Despite careful decontamination of the ice samples prior to melting, some cleaning of the data is still necessary. Often, very short-duration, high concentration spikes can be seen in the data signal due to occasional air bubbles passing through the
measurement cells in the instruments despite the use of debubbler and gas permeable membranes (accurel) immediately prior to each detector. These particular signals are easily identified and removed from the record by a simple smoothing, or (as implemented here) applying a filter that applies a threshold cutoff to the differential of the signal (due to the characteristics of these features).

Contamination signals at core breaks, due to contamination from drill fluid (Estisol-140) or general laboratory contamination,
are also relatively easily removed (Erhardt et al., 2022). This type of contamination is characterized by a steep increase in
particle count followed by an exponential decay as the contaminated sample passes through the system. For this record, the
data coinciding with signals deemed to be caused by this type of contamination have been manually removed from the dataset.

Due to the very low concentrations in Antarctic ice of all species measured and the sensitivity of instrumentation, some
measurements fall very close to or below the limit of detection of the system. Data that fall below baseline values have been
removed from the dataset.

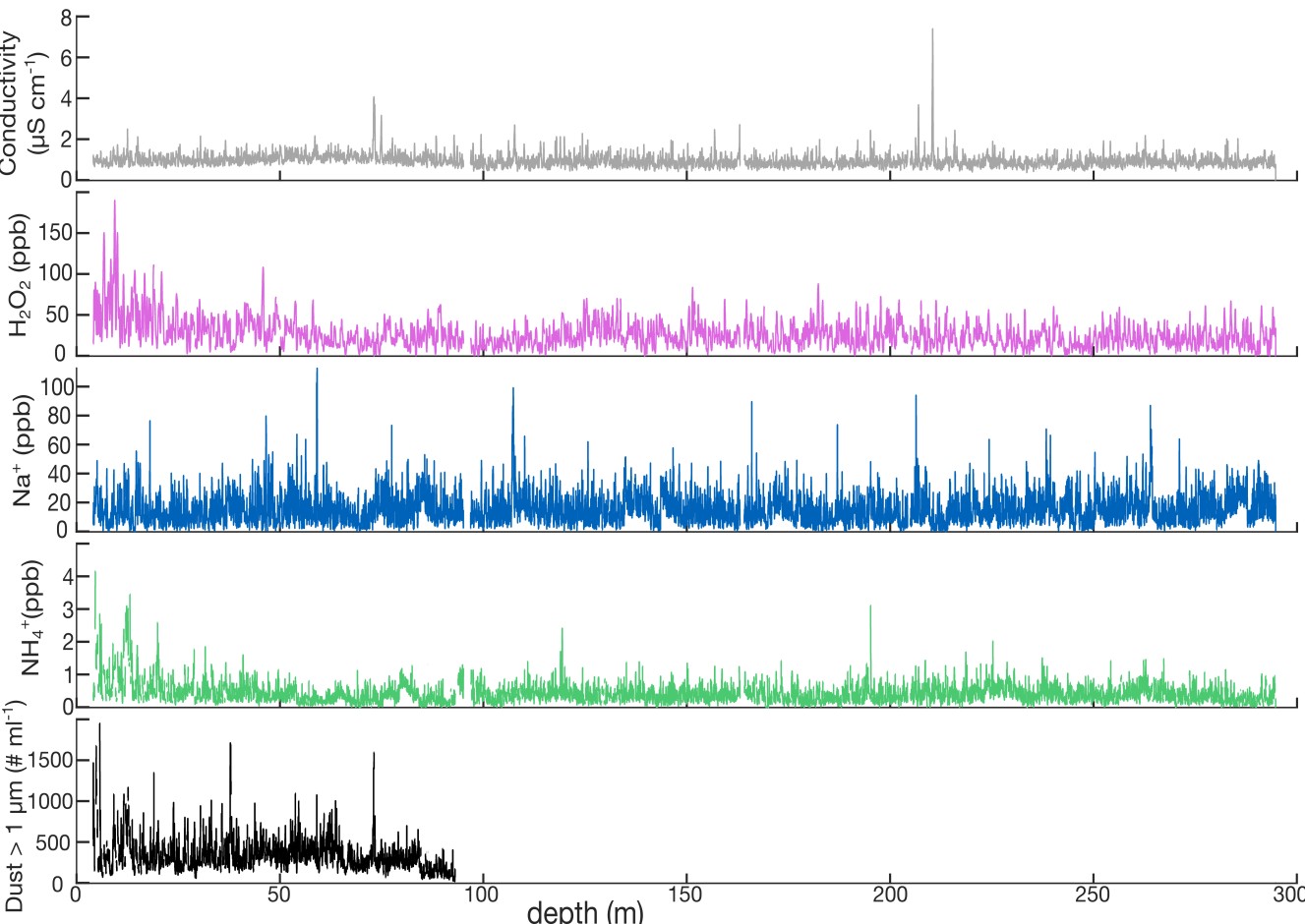

**Figure 5.** Overview of the full CFA record for MBS at 3 cm resolution. Data include electrolytic conductivity of meltwater (Conductivity,
µS cm$^{-1}$), peroxide (H$_2$O$_2$, ppb), sodium (Na$^+$, ppb), (d) ammonium (NH$_4^+$, ppb), and (e) insoluble microparticles (dust, particles per ml
>1µm). Gaps in the dataset indicate sections where ice was removed for decontamination, lost in processing, or where analytical issues
resulted in poor data quality.



**Figure 6.** 50 year subset of the 3 cm averaged resolution dataset, covering the period from 1800 to 1850 CE, plotted on the MBS2023 chronology (Vance et al., 2024b). Data shown correspond with the depth range covering approximately 77.55-63.60 m of the data presented in Fig. 5. Note the peaks in both conductivity and insoluble microparticles (dust) corresponding with the influence of the 1815 Tambora eruption, one of the volcanic horizons utilized in the development of the MBS2023 chronology (see Fig. A2).

## 7    Dataset description

We present the CFA records in three formats. First, we present the data at 1 mm resolution. Although this is in reality an oversampling of the data with regards to the smoothing inherent in the system, we consider this the full dataset at a minimum resolution allowing reduction of instrument noise. We also present the data on 3 cm resolution (3 cm averaged), which is above the effective resolution of the instrument-smoothed data for the majority of the measured species. Due to the large annual layer thickness throughout the full record (0.3 m IE yr$^{-1}$ mean accumulation with negligible layer thinning throughout the 295 m core depth), data resolution of 3 cm is able to resolve sub-annual features throughout, as seen also in the 3 cm discrete


samples (Crockart et al., 2021; Vance et al., 2024a). Figure 5 shows the full length of the CFA record on 3 cm resolution, while Fig. 6 highlights a 50-year subset of the data to demonstrate the resolution of the 3 cm averaged dataset, plotted on the
MBS2023 chronology (Vance et al., 2024b). Finally, for ease of use in climate reconstruction application, we present the record as a decadal-scale (10-year) record. The decadal scale record is computed as a 10-year mean. This is based on the MBS2023 chronology presented in Vance et al. (2024a)), and any future changes in the MBS chronology will not be reflected in this dataset.

Due to the characteristics of accumulation/deposition at the MBS site, care should be taken with interpretation of sub-annual
features produced from the 1 mm and 3 cm datasets. We refer readers to Jackson et al. (2023) for a discussion of the impacts of the intermittent precipitation and accumulation variability of the site, and Crockart et al. (2021) for satellite-era climatology and precipitation estimates.

The data presented here (Harlan et al., 2024) has been cleaned and corrected, both manually and through automated processes. Despite these extensive quality control methods, we cannot guarantee the absence of any spurious data or false signals
arising from measurement error or lab contamination.

## 8 Conclusions

Here we present a high-resolution chemistry and impurity record from the MBS ice core (MBS1718). We present records of electrolytic conductivity, insoluble microparticles (dust), calcium ($Ca^{2+}$), ammonium ($NH_4^+$), sodium ($Na^+$), and hydrogen peroxide ($H_2O_2$). Based on the chronology presented in Vance et al. (2024a), this record spans 1137 years (873-2009 CE). The
data set described here represents the highest resolution chemistry record of the full length of the MBS South ice core. We are pleased to make this dataset available for public use with considerations to the above comments on resolution and data quality, and remain available for questions by end users.

## 9 Data availability

The datasets described here (1 mm averaged, 3 cm averaged, and decadal averages on the MBS2023 age scale) are available for
download from the Australian Antarctic Data Centre (AADC) http://dx.doi.org/doi:10.26179/9tke-0s16 (Harlan et al., 2024).

*Author contributions.* The Copenhagen CFA system presented here was developed and refined by HAK and Paul Vallelonga. MH, HAK, AdC, TB, VG, SJ, and CP participated in the setup and operation of the CFA system for chemistry, melter, and discrete sample distribution. TRV participated in the MBS1718 ice core drilling. AS, HAK, and AS, CP, and TV participated in freezer work to prepare the ice core. MH, VG, TRV, and SJ constructed the depth scale. MH and AdC conducted the chemistry calibrations with the supervision of HAK. MH and
HAK wrote the initial manuscript draft. All authors have reviewed the manuscript and figures and contributed to the finalized manuscript.





*Competing interests.* The authors declare no competing interests.

*Acknowledgements.* We would like to acknowledge significant contributions from Paul Vallelonga. The authors would like to thank all participants in the 2018 and 2019 CFA campaigns, including, but not limited to Marius Simonsen, Alexander Zhuravlev, Mirjam Laderach, Nicholas Robles, Estelle Ngoumtsa, Michelle Shu-Ting Lee, Janani Venkatesh, Danielle Udy, Zurine Yoldi, Jia-mei Lin, Anna-Marie

Klüssendorf, Andrew Moy, Andy Menking, Todd Sowers, Christo Buziert, Jesper Liisberg, and David Soestmeyer. The Mount Brown South ice core project is led by Tessa Vance, and would not be possible without the team of scientists, technicians, and ice core drillers, including Paul Vallelonga, Jason Roberts, Nerilie Abram, Meredith Nation, Chelsea Long and Alison Criscitiello.

This work was supported by the Australian Government's Antarctic Science Collaboration Initiative (ASCI000002) through funding to the Australian Antarctic Program Partnership. This work contributes to Australian Research Council Discovery Project no. DP220100606.

Support for logistics and analytical funding for MBS comes from an Australian Antarctic Science grant (AAS 4414), the Australian Antarctic Division, the Carlsberg Foundation, and a European Union Horizon 2020 research and innovation grant (TiPES, H2020 grant no. 820970). HAK is supported by funding from the Novo Nordic Foundation (PRECISE), the Danish Research Foundation (1131-00007B) and H2020 (820970, 101184070). VG is supported by funding from the Villum Foundation (Villum Fonden Grant 00022995, 00028061) and the Danish Research Foundation (Grant 10.46540/2032-00228B).

**Appendix A: Supplementary tables**

**Table A1.** Recipes used in standard solutions. Standards concentrations used are shown in bold.

| | Standard Stock ($\mu$l) | Milli-Q water (ml) | Concentration (ppb) |
|---|---|---|---|
| **Multielement (Ca$^{2}$+, NH$_4^+$, and Na$^+$)** | | | |
| Stock solution | - | - | $10^5$ |
| Std 1 (first dilution) | 20 | 200 | **9.9** |
| Std 2 (first dilution) | 50 | 200 | **24.6** |
| Std 3 (first dilution) | 200 | 200 | **98.5** |
| **H$_2$O$_2$** | | | |
| Stock solution | - | - | $3\times10^8$ |
| First dilution | 30 | 100 | $9\times10^4$ |
| Std 1 (second dilution) | 100 | 75 | **120.0** |
| Std 2 (second dilution) | 50 | 75 | **60.0** |



**Table A2.** Proposed eruption events seen in MBS from Vance et al. (2024a). Events matched with CFA conductivity peaks identified as more than 3 sigma above the 3 year mean. Vance et al. (2024a) depths identified using estimation of the non-sea-salt component of the sulfate from the discrete ion chromatography measurements performed at the Institute for Marine and Antarctic Studies in Hobart, Tasmania, Australia. Dates provided are the volcanic horizons used in development of the MBS2023 chronology (Vance et al., 2024b).

| Proposed Event | MBS Depth (m) | Year (MBS2023) | CFA conductivity peak depth (m) (this study) |
|---|---|---|---|
| Pinatubo | 12.50 | 1992.9 | 12.55 |
| Agung | 25.39 | 1965.1 | 25.39 |
| Krakatoa | 53.96 | 1885.0 | - |
| Makian | 59.67 | 1864.0 | - |
| Cosiguina | 67.39 | 1837.0 | - |
| Unknown | 68.55 | 1832.0 | - |
| Galunggung | 70.91 | 1824.3 | - |
| Tambora | 73.24 | 1816.3 | 73.24 |
| Unknown | 74.88 | 1810.3 | 74.94 |
| Unknown | 86.49 | 1763.2 | 86.35 |
| Unknown | 104.32 | 1695.9 | - |
| Gamkonora | 109.15 | 1675.0 | - |
| Unknown | 114.09 | 1655.5 | 114.09 |
| Parker Peak | 117.83 | 1642.4 | 117.67 |
| Unknown | 123.72 | 1621.1 | 124.37* |
| Huaynaputina | 129.24 | 1601.6 | - |
| Ruiz | 131.15 | 1596.0 | 131.21 |
| Kuwae | 162.96 | 1459.4 | 162.96 |
| Unknown | 180.79 | 1390.0 | - |
| Unknown | 182.58 | 1382.1 | 182.59 |
| Unknown | 192.02 | 1345.8 | 192.09 |
| Unknown | 205.02 | 1287.3 | - |
| Unknown | 206.88 | 1277.5 | 207.07 |
| Unknown | 208.46 | 1269.5 | - |
| Samalas | 210.46 | 1258.3 | 210.46 |
| Unknown | 213.58 | 1241.9 | |
| Unknown | 215.64 | 1230.8 | 215.94 |
| Unknown | 223.55 | 1192.0 | - |
| Unknown | 227.58 | 1172.3 | - |
| Unknown | 240.23 | 1110.2 | 240.68* |
| Unknown | 246.36 | 1082.2 | - |
| Unknown | 255.58 | 1040.6 | - |

\* conductivity peaks with depths close to identified eruption depths in MBS, but not within 30 cm (∼1 year).



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
