# Peer review of "High resolution continuous flow analysis impurity data from the Mount Brown South ice core, East Antarctica"

_Earth System Science Data, 2024_

## Author Comment (AC2)

**Author comments:**
*We thank the reviewer for taking the time review the manuscript and provide thoughtful and constructive feedback. Reviewer comments and suggestions have been addressed; please see the below replies in blue italicized text.*

**Reply to RC2 (Anonymous reviewer):**

The Harlan et al. paper is the latest in a series of publications on the new MBS ice core, one of the rare high-resolution records of the last millennial. Harlan et al. present the CFA dataset, including meltwater electrolytic conductivity, sodium (Na+), ammonium (NH4+), hydrogen peroxide (H2O2), and insoluble microparticle measurements.

Despite some language inconsistencies, the paper is well written and easy to read. Figures have been tested for several types of colorblindness and seem adequate in terms of colors used. The quality of the figures is good and the coherent use of colors in the different figures is appreciable.

However, I have a few concerns, especially about the lack of discussion about errors and data quality assessment, as well as a proper description of the interest and novelty of such a long and highly resolved record. The different major and minor comments are discussed below.

*We thank the reviewer for acknowledging the value of our work, and for providing comments which will improve the manuscript.*

Major comments:

The dataset presented here is unique, with fine resolution and spanning a millennium. However, I feel that a key point is missing, namely the emphasis on what could be done with these new records, as well as highlighting the importance of such longer time records. Readers and future users would need information on how valuable the dataset is and on its potential future use.

*Thank you for highlighting the significance of the MBS CFA record. We have highlighted in the introduction (lines 31-34) some of the exciting work that has been published so far using the MBS records. We intentionally refrained from including further descriptive interpretation of the record, in regard to proxy interpretation and potential applications of the data for climate reconstruction, as the defined scope of the ESSD data descriptor manuscript type does not include interpretation or analysis of data presented. We do agree, however, that highlighting potential use cases would help to highlight the importance of this type of long, high-resolution climate record.*

This is particularly important as the actual minimum resolution of this dataset is considered to be 3 cm, the same resolution as the discrete (complementary) dataset published along the paper of Vance et al. (2024a). This also raises the question: what is the value, and the reliability, of a dataset with a resolution of 1 mm? A topic only partly addressed by the authors.

*Thank you for raising these concerns. We agree that this could be further clarified in the text. We will add text to the data resolution section which addresses the following points:*
- *While we present the 3 cm dataset as the "minimum resolution" dataset, our language was perhaps unintentionally misleading. While the effective smoothing of the peroxide dataset from the 2018 campaign results in an effective resolution of 3 cm, every other species has lower resolution, including at or below 1 cm in the case of the conductivity*

*signal. Resolution of 1 cm is a substantial improvement over what could be reasonably achieved with discrete sampling of a core of this length.*

- *In an effort to provide a concise and user-friendly dataset, we chose to provide all analytes on the same resolution (3 cm), rather than providing each analyte on the corresponding minimum depth resolution for that analyte (e.g. 1 cm for conductivity, 2.5 cm for NH4, 3 cm for H2O2), despite the fact that this underrepresents the resolution of the higher-resolution species like conductivity. Additionally, providing both a conservatively smoothed 3 cm dataset together with the 1 mm resolution dataset allows users to choose the best resolution for their particular uses.*

- *While the 1 mm resolution dataset is below the "actual" resolution of the dataset when accounting for data smoothing, this high-resolution dataset can be helpful in identifying extreme events in the dataset. Because the 3 cm averaged resolution dataset may smooth extreme events by reducing the amplitude and producing an artificially elevated baseline signal in the vicinity of extremes, it can be useful to provide the higher resolution dataset as a point of reference.*

On line 31, it is confusing to read a mention of ice cores in the plural when the paragraph begins with the presentation of the new Mount Brown South ice core in the singular (line 29). I think it would be interesting to first present the site and the multiple cores (currently section 1.2) and then introduce Continuous Flow Analysis (currently section 1.1). This would also provide a more logical sequence for the rest of the content. In addition, the sub-section presenting these different cores could be expanded, typically by adding information on the length and position of Alpha, Bravo and Charlie cores in relation to the main core. Consideration should also be given to adding a paragraph (or a dedicated sub-section) on what has already been done on these different cores, and possibly summarizing the main findings of Crockart et al. (2021), Jackson et al. (2023) and Vance et al. (2024a). This would also provide more clarity on the addition of this paper compared to the other publications, particularly compared to the Vance et al. paper (2024a) which already mentions the CFA data (even if it's not clear whether Vance et al. use it or not). Finally, section 4.2 on MBS chronology could be added to this part of the text instead of being presented in the data processing section (while it has not been processed in this paper).

*Many good points are raised in this comment, which we will address individually point-by-point below:*

- *It is confusing to read a mention of ice cores in the plural when the paragraph begins with the presentation of the new Mount Brown South ice core in the singular (line 29).*

  *We agree that we worded this in our initial manuscript in a confusing way. We will clarify in the text when we are referring to Mount Brown South (the ice core site), the Mount Brown South ice core array (ice cores plural), and the Mount Brown South ice core (the MBS-Main core specifically).*

- *It would be interesting to first present the site and the multiple cores (currently section 1.2) and then introduce Continuous Flow Analysis (currently section 1.1)... In addition, the sub-section presenting these different cores could be expanded, typically by adding information on the length and position of Alpha, Bravo and Charlie cores in relation to the main core.*

  *We will revisit these sections and introduce the ice core site (currently section 1.2) before introducing continuous flow analysis. We will also briefly introduce the Alpha, Bravo, and*

*Charlie shallow cores, referring the reader to the MBS2023 chronology paper (Vance et al., 2024a) for further information, as we do not present any data from these cores in this manuscript.*

- *Consideration should also be given to adding a paragraph (or a dedicated sub-section) on what has already been done on these different cores*

  *In lines 31-34 of the introduction, we briefly introduce some of the other works based on the MBS ice cores. As this manuscript is a data descriptor, and discussion of interpretation of other studies performed on the MBS ice cores falls outside the scope of a description of the CFA impurity record. Unless indicated otherwise by the editorial team, we would prefer to refrain from further discussions of other studies in this work.*

- *...particularly compared to the Vance et al. paper (2024a) which already mentions the CFA data (even if it's not clear whether Vance et al. use it or not).*

  *The MBS2023 chronology presented by Vance et al. (2024a) was developed prior to and independently of the production of the CFA impurity record presented here, and relied on stable water isotopes and discrete ion chemistry (as stated in l.168-169). We will clarify in these lines that MBS2023 did not use the CFA impurity record, and that this dataset is analytically independent from the data presented in Vance et al. (2024a) and the discrete ion chemistry dataset (Vance et al., 2024b).*

- *Finally, section 4.2 on MBS chronology could be added to this part of the text instead of being presented in the data processing section (while it has not been processed in this paper).*

  *We included section 4.2 in its current place in the text as we believe that the chronology information is most relevant in regards to the usage of the dataset (as we provide a decadally averaged dataset using this chronology), however understand that this could be confusing as the chronology was not produced as part of this manuscript/dataset. We will move the description of the MBS2023 chronology into the introduction, to follow our introduction of the MBS ice core site (what is currently section 1.2).*

Are there any traces of melt layers and wind crusts in the cores? Have their (potential) influences been considered in the various recordings? Even if it has been said that one of the criteria for selecting a site is a minimum of snowmelt in summer, it seems essential to document their occurrence and explain the potential influence of these melt layers and wind crusts on the recordings, especially for the 1 mm resolution dataset.

*In the processing of the MBS ice core(s), we have not observed melt layers and in fact have not yet documented any melt layers across the full 295 meter Main core. Certainly, in the upper 20-25 meters of the ice core, where one would logically expect a potential increase in melt due to a warming climate, we have not recorded any observations of melt, even during summer periods. The annual mean temperature of the MBS site is -27.9 degrees C, and mean summer (DJF) air temperatures are -18.4 - cold enough to preclude melt (Vance et al., 2024a).*

*Nonetheless, we have observed numerous thin (~1mm thick) ice layers in the core which have been described as "bubble free layers." These features are described in Section 2.6 of Vance et al. (2024a) and are common in Antarctic ice cores (including Law Dome and WAIS among others), and while they have been poorly studied in the past, are an active area of research in the*

*MBS cores. These features have been investigated in the Law Dome ice core in the work of Zhang et al. (2023) and are distinct from melt features. These features are also distinct from "wind crusts," but rather appear to be associated with regional scale atmospheric circulation, such as mid-latitude blocking and meridional moisture transport. Work is ongoing to investigate the chemistry and isotope signatures of these bubble free layers, and their atmospheric drivers. For the purposes of this study, the sub-millimeter scale of these features would be overcome by the internal smoothing of the CFA system, so we do not discuss them in the scope of the CFA dataset.*

My major concern is the absence of any discussion of the accuracy and precision of the measurements, which is an essential point in a data paper.

Sub-section 6.3 only presents the limit of detection (LOD), i.e. the smallest concentration that can be measured by the instrument, but not how precise and accurate these measurements are. The relative standard deviation (RSD) - the standard deviation of concentrations of the standards divided by their known levels - of the lowest level standard can be used as a measure of precision, as in Griemann et al. (2022). With regard to accuracy, intermediate standards could be considered as samples (i.e. not used for calibration) and used to define the accuracy of measurements by evaluating the difference between their known concentration and the concentration measured during the run.

*We disagree that accuracy and precision are not discussed, perhaps not using this exact wording, but throughout the text (i.e. in sections 5 and 6) we carefully list uncertainties and errors on the data that influence the final accuracy and precision. For example, in Section 4.1 on the depth scale, we discuss the uncertainties on depth scale and provide numbers on these and in section 6 we specifically discuss the analytical precision and data quality. However, in addition to this, we will add the relative standard deviations of the standards used (following Griemann et al. (2022), as suggested) alongside the LOD of the analytes in Table 2, as well as text describing this in Section 6.2.*

l.267: Have you compared the 3 cm resolution Na+ dataset with Vance et al.'s (2024a) discrete dataset? This could be used as (partial) data quality assessment.

*We agree that a comparison to the published discrete dataset would provide an interesting interlaboratory comparison, however we point out that this might be misleading if presented purely as a data quality assessment tool. The discrete measurements were analyzed using ion chromatography, and there is a similar (if not higher) likelihood of e.g. laboratory contamination for these measurements as for the CFA measurements.*

*We ultimately chose not to present a comparison to the Vance et al. (2024a) discrete dataset, as the defined scope of data description manuscripts published in ESSD states that "any comparison to other methods is beyond the scope of regular [data descriptor] articles."*

Similarly, no error is mentioned for the third dataset presenting a decadal scale record. A global error calculation - including measurement and dating errors - should be associated with the decadal scale record.

*We are unsure what the reviewer means by a "global error calculation" here. However, the errors associated with the measurements are described in the text of the manuscript, which is linked by DOI to the dataset as available from the Australian Antarctic Data Centre. Therefore, users of the decadal scale data set will have easy and open access to the descriptions in this text when accessing and using the dataset. Additionally, further discussion of layer counting error (as*

*well as a comparison to the WAIS and Law Dome ice cores) is thoroughly detailed in the manuscript accompanying the MBS2023 chronology (Vance et al., 2024a), which is referenced in the manuscript text.*

Section 4.4 on signal delay time is not detailed enough. Are the numbers and errors the result of statistical analysis? How many estimates have led to these values? How regularly were the bulk and individual delay times measured? And, in practice, how is the "total" delay time applied to the different measurements?

*We thank the reviewer for raising the matter of delay time estimation. We will include in the text more detail on the calculation of the bulk delay times based on the individual sample runs, and how many/often standard runs were used to make these calculations.*

I don't see how section 6.3 (l.256-261) can be used to verify data quality and depth scale accuracy. Firstly, it should be specified what data are involved here (conductivity) and the comparison between the conductivity measured by CFA and the discrete measurements of nssSO4 by Vance et al (2024a) should be clearly mentioned. Secondly, approximating 90 cm as equivalent to 3 years is not correct: it is mentioned in the text that accumulation is highly variable on an interannual scale. Moreover, even if the accumulation were more or less constant, we assume that the authors are using a sliding average of 90 cm on the raw data (observed depth), but density changes with depth, so 90 cm at the surface does not represent the same number of years as at a depth of 200 m when the snow has been compacted into ice. Thirdly, I disagree with the statement "we are able to identify many of the volcanoes reported in Vance et al. (2024a)". Looking at Table A2 in the Appendix, only 16 of the 32 volcanoes identified in Vance et al. (2024a) are also present in the conductivity record, i.e. 50%. This deserves an attempt at an explanation (for example, it is possible that the relative proximity of the coast induces a greater presence of impurities resulting in a higher background noise in the conductivity measurements).

*Based on these comments, we understand that this section of the text requires some clarifications. We do state in the text of section 6.3 that we are using the CFA conductivity measurements (presented in this manuscript), and comparing to volcanoes identified in Vance et al. (2024a), wherein the methods of volcanic identification are described. We will respond to the individual comments about the content of this section point by point below:*

- I don't see how section 6.3 (l.256-261) can be used to verify data quality and depth scale accuracy.

  *Our intention in this section is to demonstrate the ability of the CFA record described here to correspond with signals identified in the discrete dataset. We consider that a matching eruption signal to within one year depth uncertainty is a basic validation of the general accuracy of the depth scale used to prepare this dataset. We will rephrase the opening of section 6.3 to better describe our intentions with this matching exercise.*

- *Firstly, it should be specified what data are involved here (conductivity) and the comparison between the conductivity measured by CFA and the discrete measurements of nssSO4 by Vance et al (2024a) should be clearly mentioned.*

  *It is stated in line 256 that we use conductivity peaks above 3 $\sigma$ in this matching exercise, compared to the data provided in Vance et al. (2024a). We will add text to this section to indicate that the Vance et al., (2024a) dataset uses the non-sea salt sulfate signal to identify volcanic eruption signals.*

- *Secondly, approximating 90 cm as equivalent to 3 years is not correct: it is mentioned in the text that accumulation is highly variable on an interannual scale. Moreover, even if the accumulation were more or less constant, we assume that the authors are using a sliding average of 90 cm on the raw data (observed depth), but density changes with depth, so 90 cm at the surface does not represent the same number of years as at a depth of 200 m when the snow has been compacted into ice.*

  *We understand that 90 cm does not correspond to exactly 3 years throughout the full core, however, as we are comparing volcanic signal depths identified in the CFA conductivity to the depths of corresponding volcanic signals identified in Vance et al. (2024a), we chose to use a sliding window of fixed depth. We state in the section that this is an "approximation" of 3 years, as this is true as an average approximation across the core. As we have clearly described the accumulation variability characteristic of the site, we consider that describing this as an "approximation" is not inaccurate. We will update the text to clarify that this is indeed only an approximation and does not represent exactly three years throughout the length of the record, due to accumulation variability and density changes downcore.*

- *Thirdly, I disagree with the statement "we are able to identify many of the volcanoes reported in Vance et al. (2024a)". Looking at Table A2 in the Appendix, only 16 of the 32 volcanoes identified in Vance et al. (2024a) are also present in the conductivity record, i.e. 50%.*

  *We will rephrase the text in line 257-258 to the following: "Using this method, we are able to identify 16 of the 32 volcanic events reported in Vance et al. (2024a) to within one year age-at-depth uncertainty."*

- *This deserves an attempt at an explanation (for example, it is possible that the relative proximity of the coast induces a greater presence of impurities resulting in a higher background noise in the conductivity measurements).*

  *The reviewer is correct here, in that as we are comparing bulk conductivity in the CFA record with the discrete non-sea salt sulfate record, there will be inherent differences in the two datasets. However, we have intentionally avoided substantial interpretation of the record here, as ESSD specifies that significant data interpretation is beyond the scope of a data descriptor manuscript. However we will add the following statement to the text:*

  *"We consider the ability of the CFA conductivity signal to identify half of the volcanic events found in the non-sea salt sulfate signal by Vance et al. (2024a) across the MBS record to be an external validation of the CFA conductivity record described here. We attribute existence of peaks that are not identified both datasets to the different datasets used (comparing conductivity to non-sea salt sulfate), as it is likely that some peaks in the conductivity signal are related to sources of other soluble ions, in addition to volcanic sulfate."*

Minor comments:

Some inconsistencies in the text:

Two periods of CFA measurements are mentioned in the text but using different terms (CFA sampling/measurement/melting campaign) (l.54, l.218-219, l.238). The same wording should be used to facilitate understanding.

*We will update the text to ensure consistent wording throughout.*

The depths given for dry-drilled and wet-drilled sections are not the same throughout the text: for example, in l.95-96, dry-drilled and wet-drilled sections are respectively at ∼4-94 m and ∼95-295 m vs. 5-95 m and 96-295 m in l.218-219.

*We will update the text to ensure that the relevant depth ranges are described accurately and consistently throughout the manuscript.*

There is a problem with the table numbering. The first Table mentioned in the text is actually Table 2 (line 211). The numbering of the figures and the order in which they are presented should be changed in the text accordingly.

*We thank the reviewer for pointing out this error in table numbering. We ensure that the tables are presented in the correct order in the revised manuscript.*

l.2: In the abstract, the authors mention a mean annual accumulation estimated at 20-30 cm ice equivalent. However, 20 cm is never mentioned in the main text. The abstract should only contain information that is present in the main text.

*We will update the abstract to reflect only the information in the manuscript.*

Figure 1 (near l.61): some values of lat-lon would be welcome.

*We will update Figure 1 to include lat/lon values.*

Legend of Figure 2 (near l.80): "Shaded areas indicate cuts made with a bandsaw." Do you mean dotted lines?

*We thank the reviewer for pointing this out. There is an error in the figure caption for Figure 2. It should state "dotted lines indicate cuts made with a bandsaw. Shaded area (hatched line) indicates surfaces planed for intermediate layer core scanning." We will update the caption to correct this.*

l.90: A few more details on the scraping of horizontal ends could already be given here: at the very least, mention how much thickness is removed and insist that this is taken into account in the depth logs, especially as one of the datasets is published at millimeter scale.

*We will add text at line 91 stating the following: "1 - 2 mm of ice was removed in this cleaning process. All sample pieces were measured after cleaning, and any ice removed in cleaning is accounted for in the depth record."*

l.95: The authors should mention why the dry-drilled section only begins at a depth of 4 meters and how the chronology of these first 4 meters is carried out.

*As is common with intermediate/deep ice cores, the drilling took place from a trench dug into the firn at the ice core site. This accounts for the 4 meters missing from the top of the Main core. These 4 m were matched with the overlapping shallow cores drilled nearby during the same season. This is described in the manuscript by Vance et al. (2024a), and as we do not present the*

*data from the shallow cores or the chronology, we refer readers to Vance et al. (2024a) for this information.*

l.114: It would be interesting for the readers to mention which type of analysis will be performed in the future.

> *At time of submission, it is not known exactly which analyses will be performed in the future. As such, we consider this to be beyond the scope of this data descriptor manuscript.*

l.137: What are the resolution and accuracy of this cable-driven rotary encoder (in comparison to the previous system)?

> *The Waycon SX-80 has a sensitivity of γ = 25 counts mm−1, and linearity of +/- 0.15%. We will add text that describes the specifications of the instrument at line 137-138.*

l.145: The last sentence of the paragraph raises a question: why would we need more details about the gas extraction system if it was not used for data collection? The answer is only 3 pages later. I suggest adding a few words alluding to the effect of gas extraction on the CFA data presented here.

> *Thank you for the suggestion. For brevity and to avoid unnecessary repetition, we will amend the last sentence of the paragraph to state "Further detail on the gas extraction system and its impact on measurement quality follows in section 6.1 below."*

l.159: What is the impact of inaccuracies of one or two millimeters on the 1 mm dataset? This should be estimated and mentioned in the text.

> *The impact of inaccuracies on the scale of 1 - 2 mm are described in section 4.1 describing the depth scale. Such inaccuracies would be well below the annual resolution of the record and are unlikely to be of climatic interest to users of this dataset.*

l.165: It is mentioned that the procedure for correcting differences in measurements of the same stick is presented in Vance et al. (2024a). However, Vance et al. (2024a) state that "The scaling and shift factors will be described in detail as the CFA trace chemistry and water isotope datasets are developed and published". So it seems that information is missing here.

> *We apologise for this omission. This was a two-step process: first, the depth scale for the entire Main core needed to be developed taking into account small differences in field to lab core lengths, and then also small differences in the IC (discrete chemistry) stick lengths. This is described in detail in Vance et al. (2024a). However, as the reviewer states, the shift and scaling factors (and CFA datasets) were still under development at the time of publication of Vance et al. (2024a). These discrepancies were accounted for in the assignment of the top depths for each meter-long core section. Further description of the depth scale procedures can be found in Gkinis et al. (2024). We will update the text to more clearly describe the depth adjustment process.*

l.184: For readers unfamiliar with CFA measurements, it might be interesting to explain the principle of calibration with standards run at the beginning and end of each measurement. Is it a single calibration with some standards run at the beginning and others at the end? Or two distinct calibrations to account for e.g. measurement drifts?

> *The standards run procedure is relatively standardized across most CFA systems. In line 188, we refer to Kaufmann et al. (2008), wherein there is a very clear and concise description of the*

*standard calibration procedures, which we use also here. As this is a data descriptor manuscript, and not a methods paper, we will update the text to more specifically refer the reader to the work of Kaufmann et al. (2008).*

l.189: Shouldn't the MATLAB script mentioned be made available in a code availability section, as required by the ESSD guidelines?

*The MATLAB scripts used here only perform simple calculations in an automated manner. All computations performed in the MATLAB scripts used in data processing are described in the text of the manuscript, and as such, we do not consider the scripts themselves to provide significant additional information to the reader. Additionally, the majority of similar ESSD manuscripts (e.g. Erhardt et al. (2023)) do not include scripts used for simple data processing/calibrations.*

l.205-206: Attention should be paid to the significant figures used here (either 79 ± 4 seconds or 79.x ± 3.6 seconds).

*We thank the reviewer for pointing this out. We will update the text to ensure consistency of significant digits throughout.*

l.254: Is Figure 4 needed? It is not really discussed in the text and is quite redundant with Table 2.

*We find Figure 4 to be a helpful means of visualizing the range of variability of the dataset, however we will consider moving it to the supplementary materials if the manuscript length is a concern.*

l.279: When mentioning "baseline values", do you mean "true" observed baseline or the limit of detection? It seems more accurate to define the LOD as a threshold.

*The baseline values referred to here are the measured MilliQ (Merck RiOs™16 MilliQ) levels. We will update the text in line 279 to clarify this.*

Figure 5:

1) The last data point of each complete series reaches 0. If this is an artifact of the plotting (which it probably is), the last point of each series should be removed.

*The reviewer is correct that this is a plotting artifact. We will correct this in the revised figure.*

2) It looks like there are several points reaching 0 in the H2O2, Na+ and NH4+ series. If this is the case, these points are probably below the LOD/baseline (especially for H2O2 and Na+) and should be removed according to text line 279. It may be interesting to try to plot the LOD for these series (even though they are low for H2O2 and very low for NH4+).

*As described in the reply to the comment on Line 279, MilliQ the baseline value used was the threshold for data removal (which will be clarified in the revised text). We will add the LOD to the plot in Figure 5 to provide a visual representation of the information provided in Table 2.*

l.286: Is there a reference to confirm that the layer thinning is negligible at this ice core site? This is surprising, given that layer thinning already has a significant effect at 80 m depth in records such as that from Philippe et al. (2016). I also recommend adding the cause of this "layer thinning" (due to strain rates) to the text, to leave no doubt for the reader.

*Layer thinning is expected to be small at this site, with a bedrock depth of approximately 2000 metres, as discussed in Vance et al., 2016. However, the reviewer is correct that there will be some layer thinning due to strain across the core, and we will note this in the text. Layer thinning does not affect the use or interpretation of this dataset (which is already on an established chronology), but is of course taken into account when developing accumulation records (regardless of how small layer thinning is expected to be).*

Technical corrections

Pay attention to sentence syntax (same word used several times in the same sentence):

l.2: the second word 'accumulation' can be deleted.
l.184: twice the word 'run' in the same sentence.
l.223-224: the first part of the second sentence (this gives a smoothing effect) repeats the first sentence (the dynamics lead to smoothing).
l.226: 'calculated' used twice in the same sentence.
l.227: similar as previous comment with 'using' and 'use'.
l.272: similar as previous comment with 'applying' and 'applies'.

*We thank the reviewer for pointing these out. We will revise the text to avoid unnecessary repetition.*

Some inconsistencies in the writing:

l.99: lowercase after the ':' (as in l.7 for example).
There should always be a space between the number and the units (see ESSD guidelines): l.123, l.163, legend of Table 1, l.227, legends of Figures 4 and 5 (1 µm)
(e.g. l.152) The word 'meltrate' is sometimes written 'melt-rate'.
l.178-179: standardize the expression "a # step calibration" or "a # step calibrations".
l.218-219: interval values (in this case, depth values given in brackets) must either be joined to the dash or separated by a space, but must be consistent throughout the text (including in Table 1 or in other intervals like in l.95-96, l.139-140, legend of Table 1, l.227, …).
l.225: CFA system (with a lower-case s, for consistency).

*We thank the reviewer for identifying these typographical errors and inconsistencies. We will update each instance to ensure the text is correct and consistent with ESSD guidelines.*

Some bibliography citations need to be revised:

General question: which order do you use to cite multiple references in text (for example, in l.112-113)?
l.100: should be "(Bigler et al., 2011)".
l.122: should be "in Dallmayr et al. (2016)."

*We will cross check the references to ensure that they are presented correctly and consistently throughout the text.*

Purely technical corrections:

l.18: delete the comma before (Legrand and Mayewski, 1997).
*We will correct this.*

l.122: a point is missing after the citation.
   *We will correct this.*
l.148: "The position derived *from* melt-rate data".
   *This is correct as written, as we refer here to the melt-rate data which was derived from the position of the core at each time step (we will revise this as "position-derived melt-rate" for clarity).*
l.199: "(15 seconds for both *CFA setup* systems)".
   *We will revise the text to read "(15 seconds for both the 2018 and 2019 CFA setups)"*
l.219: meltrate (instead of metlrate).
   *We will correct this.*
l.220: the second parenthesis ) is missing at the end of the sentence.
   *We will correct this.*
Table 1: min-1 should be in superscript.
   *We will correct this.*
l.229: min-1 should be in superscript.
   *We will correct this.*
l.254: a point is missing at the end of the sentence.
   *We will correct this.*
Legends Figures 4 and 5: perhaps the expression should be "particles > 1 µm per ml".
   *We will update the figures to use this phrasing.*
l.272: I suggest adding a 'by' before 'applying'.
   *We will correct this.*

Table A2: a hyphen is missing at 213.58 m.
   *We will correct this.*

References:

Grieman MM, Hoffmann HM, Humby JD, et al. Continuous flow analysis methods for sodium, magnesium and calcium detection in the Skytrain ice core. Journal of Glaciology. 2022;68(267):90-100. doi:10.1017/jog.2021.75

Philippe, M., Tison, J.-L., Fjøsne, K., Hubbard, B., Kjær, H. A., Lenaerts, J.T.M., Drews, R., Sheldon, S.G., DeBondt, K., Claeys, P., and Pattyn, F.: Ice core evidence for a 20th century increase in surface mass balance in coastal Dronning Maud Land, East Antarctica, The Cryosphere, 10, 2501–2516, https://doi.org/10.5194/tc-10-2501-2016, 2016.

*References:*

*Erhardt, T., Jensen, C. M., Adolphi, F., Kjær, H. A., Dallmayr, R., Twarloh, B., Behrens, M., Hirabayashi, M., Fukuda, K., Ogata, J., Burgay, F., Scoto, F., Crotti, I., Spagnesi, A., Maffezzoli, N., Segato, D., Paleari, C., Mekhaldi, F., Muscheler, R., Darfeuil, S., and Fischer, H.: High resolution aerosol data from the top 3.8 ka of the EGRIP ice core, Earth System Science Data Discussions, 2023, 1–21, https://doi.org/10.5194/essd-2023-176, 2023.*

*Gkinis, V., Jackson, S., Abram, N.J. et al. An East Antarctic, sub-annual resolution water isotope record from the Mount Brown South Ice core. Sci Data 11, 986 (2024). https://doi.org/10.1038/s41597-024-03751-w*

Grieman M. M., Hoffmann H. M., Humby J. D., et al. Continuous flow analysis methods for sodium, magnesium and calcium detection in the Skytrain ice core. Journal of Glaciology, 68(267):90-100. doi:10.1017/jog.2021.75, 2022.

Vance, T. R., Abram, N. J., Criscitiello, A. S., Crockart, C. K., DeCampo, A., Favier, V., Gkinis, V., Harlan, M., Jackson, S. L., Kjær, H. A., Long, C. A., Nation, M. K., Plummer, C. T., Segato, D., Spolaor, A., and Vallelonga, P. T.: An annually resolved chronology for the Mount Brown South ice cores, East Antarctica, Climate of the Past, 20, 969–990, https://doi.org/10.5194/cp-20-969-2024, 2024a.

Vance, T. R., Abram, N. J., Gkinis, V., Harlan, M., Jackson, S., Plummer, C., Segato, D., Spolaor, A., Vallelonga, P., Nation, M. K., Long, C., and Kjær, H. A.: MBS2023 - The Mount Brown South ice core chronologies and chemistry data, Ver. 1, Australian Antarctic Data Centre, https://doi.org/doi:10.26179/352b-6298, accessed: 2024-09-07, 2024b.

Zhang, L., Vance, T. R., Fraser, A. D., Jong, L. M., Thompson, S. S., Criscitiello, A. S., and Abram, N. J.: Identifying atmospheric processes favouring the formation of bubble-free layers in the Law Dome ice core, East Antarctica, The Cryosphere, 17, 5155– 5173, https://doi.org/10.5194/tc-17-5155-2023, 2023.

---

## Author Response (AR2)

**Manuscript response file for ESSD-2024-335 "High resolution continuous flow analysis impurity data from the Mount Brown South ice core, East Antarctica"**

**Included in this file:**

- (1) Overview of changes made to manuscript.
- (2) Authors' reply to topic editor discussion (in *red italicized text*)
- (3) Comments from referees in black text.
- (4) Authors' response (as a point-by-point reply to referee comments, updated from the discussion stage of the submission, included here in *blue italicized text*).
- (5) Authors' changes in manuscript (in *red italicized text* following the author responses. Additional changed made to the text described following the reply to reviewers).
- (6) Tracked-changes document produced using latexdiff.

**Overview of major changes made to manuscript:**

- Changes made to manuscript in response to referee comments, as described below in point-by-point reply to reviewers.
- Some sections of the manuscript have been re-arranged for clarity and readability of the manuscript mainly in the introduction. This is reflected in the tracked-changes document.
- A table has been added (now Table 1) with the delay time for each analyte measured from the standards.
- The order of what are now Tables 2 and 3 have been swapped so that they match the order in which they are referenced in the text.
- In a final inspection of the full dataset, the flowmeter data from the final run was found to be faulty, impacting only the insoluble microparticle data below approximately 85 m depth. This data has been removed from the published dataset as well as Figure 5 as we are not able to confirm its accuracy.
- MATLAB data calibration script has been included as a supplementary file.

**Reply to Topic editor decision**

Public justification (visible to the public if the article is accepted and published): Thank you for sharing the MATLAB code. PDF is not a great format for source code. I'll let this go because it's such a short script, but in general you should provide an ASCII file - with the data or in a new Zenodo repository.

Code has been re-submitted as a .txt file in ASCII format.

I know that in an earlier email I wrote that I am supportive of your replies to reviewers and welcome a revised paper. Having read the revised paper, and the response to reviewers again, I have decided to ask for a few more changes as requested by the reviewer. R2 had some good "major" points that they raised and your declined to address them. Some of these were around Vance (2024). Part of your reply decision may be that Vance is a co-author on this work and therefore I assume you are very familiar with their work. Readers of your ms and users of this data are not likely to have that same familiarity.

Manuscript has been revised to address these reviewer comments – see further explanations below.

In response to R2 suggestion "l.267: Have you compared the 3 cm resolution Na+ dataset with Vance et al.'s (2024a) discrete dataset? This could be used as (partial) data quality assessment.", you cite https://www.earth-system-science-data.net/about/aims\_and\_scope.html and "Any comparison to other methods is beyond the scope of regular articles." I disagree with your use of this clause as justification for not addressing the reviewer comments. This clause is about methods. The request is to compare data, not methods. Differences in the data may be attributable to methods, and you do not need to dig into Vance methods to try to explain differences. But differences in the data are likely to be helpful to readers who may want to use your data. ESSD products should have robust uncertainty, and validation. It is challenging to do a good validation in data-sparse Antarctica, so comparing against similar products and honestly discussing differences is often the best we can do.

Thank you for providing further clarity on what is/is not within the scope for a data descriptor. We have revised the manuscript to include a comparison with the discrete sodium data from the Vance et al., 2024 dataset, presented in a new section (6.2). We present a visual comparison in an update to Figure 6, as well as a brief discussion and statistical comparison of the two datasets (lines 375-397).

In response to your p11 reply to R2 where you write "However, we have intentionally avoided substantial interpretation of the record here, as ESSD specifies that significant data interpretation is beyond the scope of a data descriptor manuscript." -- I appreciate your adherence to the ESSD guidelines. I am almost always telling people to remove analysis and interpretation that focuses on what I think of as 'downstream science'. I have never had to encourage authors to add interpretation and analysis. In your case, I think you are taking the ESSD advice too literally. It is important to analyze and interpret your data. Here R2 is asking you to do this to help people interpret the quality of the data, what impurities might be present, signal to noise ratio, etc.

It would be out of scope to interpret ash layers and start using this to quantify, say, atmospheric sulfur injection from volcanic events, or what volcano locations that are or are not captured in the record means for changing wind patterns. But it would be in scope to discuss how changing wind patterns might influence which eruption events are captured in the data, when the goal of that discussion is to help readers understand limitations of the data.

We have expanded section 6.1 to further discuss the potential causes of the differences in the ability of the two datasets to identify volcanic peaks (lines 352-373).

Overall, the ms is much improved by the reviewer suggestions and your updates. I would like to see some

additional work responding to R2 'major' suggestions highlighted above, which seem to center around Vance. I hope that given your familiarity with that work, and their contributions to this work, that this is an easy change. If you can make these changes I would be happy to send a revised manuscript back to reviewers.

**Other comments:**

In response to R2 you write, "effective resolution of 3 cm, every other species has lower resolution, including at or below 1 cm". I often confuse myself here, but I believe 1 cm is higher resolution than 3 cm, and I note two paragraphs below you refer to 1 mm resolution as "high".

Thank you for catching this mistake – it has been corrected for clarity in the reply document below.

L332: "potentially using a lower deviation threshold value" I just said you don't need to address methods differences and could just focus no data, but nonetheless this vagueness is surprising given that Vance is a co-author on this paper and the lead author of this paper is a co-author on the Vance paper. If you can concretely and succinctly explain why you see a different number of volcanoes, even if it does mean discussing methods, that would not be a bad thing. Vance either did or did not have a lower deviation threshold.

The text has been updated in section 6.1 to be more specific about the methods used in the Vance et al., 2024 manuscript, and how they compare to the work presented in this manuscript.

Thank you,

Ken Mankoff

**Author comments:**

We thank the reviewer for taking the time review the manuscript and provide thoughtful and constructive feedback. Reviewer comments and suggestions have been addressed; please see the below replies in blue italicized text.

Changes made to the revised manuscript in red italicized text.

**Reply to RC1 (Tobias Erhardt):**

Harlan et al. present a description of the measurement setup for the MBS CFA record and the associated uncertainties. Overall I found the manuscript to be complete, well written and very readable!

We thank the reviewer for the thoughtful feedback.

I only have some minor remarks that the authors should address before publication:

The most important one is a potential issue with the relative delay estimation between the two parameters (Section 4.4): The authors determine the signal delays using the maximum of the derivative during the transition into the standard measurements. This approach is certainly valid, but a small detail is missing from the discussion: The influence of the signal smoothing on this way of determining the relative signal delays. For higher signal smoothing the maximum of the derivative will occur systematically later as the derivative will have its maximum at approximately halve the signal amplitude. This will introduce a slight systematic overestimation of the signal delay for parameters with larger smoothing relative to parameters with less signal smoothing. I would encourage the authors to add a few sentences in this regard.

We thank the reviewer for bringing this up - we agree that this is an important point and well raised. We chose this method due to its ability to be applied to each run in a systematic way but agree that (as with all approaches) there is a certain degree of bias dependent on signal smoothing inherent to this method. We will text to address this:

"Other choices for delay calculations could have been to use the start of the rise or even the end of the rise, however due to smoothing can be hard to reliably identify, whereas the maximum of the derivative of the increase is easily and systematically identified. However, we recognize that this can introduce a minor offset between methods that are highly smoothed in comparison to those that have faster response times."

Changes made in manuscript: Text added in section "Signal delay time," (lines 243-266) as described above.

The second is the description of the differences between the 2018 and 2019 setups. Judging from the flow diagrams in Figure 3, the systems were quite different with respect to the parameters presented here. I would suggest the authors expand this section significantly to delineate all the differences also in terms of procedures.

We agree that more detail could be warranted by other CFA users/readers, especially with regards to run procedures. The most meaningful differences in the two system setups are described in the manuscript, namely the depth registration types i.e. the difference between using an encoder and a laser. We see the point raised that the two instruments/methods could influence the final depths slightly differently, however these effect of these differences on the

final dataset are likely very minor. Some of the potential differences between the two methods include the shorter "dead time" with the laser vs the cable encoder, which must be fully removed and replaced during frame changeover. Additionally, while the 55 cm piece lengths used in the 2018 campaign introduce more potential contamination at breaks, they make absolute depths more certain, as the top depths are known at every 55 cm rather than every meter. We will add a few more lines regarding that in the section titled "Depth scale".

Besides these depth differences, the systems were not in fact too different in the end (i.e. delay time to the conductivity measurement was in both systems 15 seconds (stated in section 4.4). Calibrations and processing of calibrations were made based on the same concentration and dilution of standards, following the same calibration data processing method. In section sampling resolution we explain how the different melt-rates influenced the resolution "These melt-rates produce a direct sampling resolution (independent of smoothing due to response time) of 0.58 and 0.68 mm, respectively"

We will add text to the methods sections specifically delineating the elements of the systems and procedures that were the same and/or different, elaborating on the system diagrams in Figure 3.

New section added "2.3 Primary differences between 2018 and 2019 CFA setups" on what is now page 8. This section details the most significant changes that were made to the system, and describes the anticipated impact of these changes on the data quality.

**Specific remarks:**

L 123: "temperamental" is an odd choice of words. To avoid personification maybe describe in terms of noise sensitivity

We will update the text with more appropriate terms such as 'inconsistent' or 'erratic'.

The proposed change has been made in revised manuscript.

Section 3.0.2: See comment above.

Section 4.4: See comment above.

Section 5.0.1: Explain target meltrate especially in light of the density change downcore.

While we present here the melt-rate throughout each measurement campaign, during the 2018 campaign (top ~95 m), the melt-rate did in fact decrease as the firn/ice density increased downcore. We will elaborate further on this in section 5.0.1 to provide more clarity on how the melt-rate was adapted to the density changes.

Further detail on meltrate changes with respect to firn densification downcore have been added to section 4.1 (lines 279-281).

Section 6.1: What makes the pH measurements more sensitive to the pressure fluctuations than the Na channel? From my understanding both are absorption spectroscopic methods and are thus more sensitive to mixing ratio fluctuations than the fluorescence detection.

Indeed both absorption methods are more sensitive than fluorescence methods. The acid

method however is even more sensitive than the sodium method, because rather than being a "true" absorption method it relies on pH dye changes using a combination of two dyes to cover the full range of pH expected in ice core waters One could thus think of it as a double "absorption" function, the dyes are each most efficient at one wavelength, but do impact each others max wavelength. The method is optimized and described in Kjær et al. (2016) so we refer to that for more details. Because the method is very sensitive to the dye to sample ratio, small flow changes (i.e. squeezing and wear of pump tubing, which naturally happens over time, as well as pressure changes and temperature changes) influence the method more than the also sensitive Na+ method. In the Na+ method, the issue (more so than exact mixing of reagent and standards) seems to be due to wear out of the IMER column over time, causing a slower drift. We refer to Kaufmann et al. (2008) for more information on that method.

Section 6.5: Please explain the filtering approach detailed enough so that others are able to replicate the same thing, if needed. This will also help to judge the imprint of the filter on the final dataset in terms of its frequency content.

We state in section 6.5 that the filtering approach was "applying a filter that applies a threshold cutoff to the differential of the signal (due to the characteristics of these features)." We agree that this could use more detailed elaboration. Text will be added to the manuscript which more explicitly describes the method of data filtering, stating text along the lines of the following:

"The passage of air bubbles through the detectors results in a signal characterized by a significantly steeper increase (and subsequent decrease) in the signal voltage than anything produced by variability in the ice core sample. Because of this characteristic peak shape, we are able to define for the differential of the dataset, a threshold, values above which we use to define "bubble spikes" in the data, which can be safely removed from the dataset."

Text has been added to section 5.5 to further explain the filtering approach used for data cleaning (lines 328-330).

**References:**

Kaufmann, P., Federer, U., Hutterli, M. A., Bigler, M., Schüpbach, S., Ruth, U., Schmitt, J., and Stocker, T. F.: An improved continuous flow analysis system for high-resolution field measurements on ice cores, Environmental Science & Technology, 42, 8044—8050,https://doi.org/10.1021/es8007722, 2008.

Kjær, H. A., Vallelonga, P., Svensson, A., Elleskov L. Kristensen, M., Tibuleac, C., Winstrup, M., and Kipfstuhl, S.: An Optical Dye Method for Continuous Determination of Acidity in Ice Cores, Environmental Science & Technology, 50, 10 485–10 493,https://doi.org/10.1021/acs.est.6b00026, pMID: 27580680, 2016.

**Author comments:**

We thank the reviewer for taking the time review the manuscript and provide thoughtful and constructive feedback. Reviewer comments and suggestions have been addressed; please see the below replies in blue italicized text.

Changes made to the revised manuscript in red italicized text.

**Reply to RC2 (Anonymous reviewer):**

The Harlan et al. paper is the latest in a series of publications on the new MBS ice core, one of the rare high-resolution records of the last millennial. Harlan et al. present the CFA dataset, including meltwater electrolytic conductivity, sodium (Na+), ammonium (NH4+), hydrogen peroxide (H2O2), and insoluble microparticle measurements.

Despite some language inconsistencies, the paper is well written and easy to read. Figures have been tested for several types of colorblindness and seem adequate in terms of colors used. The quality of the figures is good and the coherent use of colors in the different figures is appreciable.

However, I have a few concerns, especially about the lack of discussion about errors and data quality assessment, as well as a proper description of the interest and novelty of such a long and highly resolved record. The different major and minor comments are discussed below.

We thank the reviewer for acknowledging the value of our work, and for providing comments which will improve the manuscript.

**Major comments:**

The dataset presented here is unique, with fine resolution and spanning a millennium. However, I feel that a key point is missing, namely the emphasis on what could be done with these new records, as well as highlighting the importance of such longer time records. Readers and future users would need information on how valuable the dataset is and on its potential future use.

Thank you for highlighting the significance of the MBS CFA record. We have highlighted in the introduction (lines 31-34) some of the exciting work that has been published so far using the MBS records. We intentionally refrained from including further descriptive interpretation of the record, in regard to proxy interpretation and potential applications of the data for climate reconstruction, as the defined scope of the ESSD data descriptor manuscript type does not include interpretation or analysis of data presented. We do agree, however, that highlighting potential use cases would help to highlight the importance of this type of long, high-resolution climate record.

This is particularly important as the actual minimum resolution of this dataset is considered to be 3 cm, the same resolution as the discrete (complementary) dataset published along the paper of Vance et al. (2024a). This also raises the question: what is the value, and the reliability, of a dataset with a resolution of 1 mm? A topic only partly addressed by the authors.

Thank you for raising these concerns. We agree that this could be further clarified in the text. We will add text to the data resolution section which addresses the following points:

• While we present the 3 cm dataset as the "minimum resolution" dataset, our language was perhaps unintentionally misleading. While the effective smoothing of

the peroxide dataset from the 2018 campaign results in an effective resolution of 3 cm, every other species has higher resolution, including at or below 1 cm in the case of the conductivity signal. Resolution of 1 cm is a substantial improvement over what could be reasonably achieved with discrete sampling of a core of this length.

- In an effort to provide a concise and user-friendly dataset, we chose to provide all analytes on the same resolution (3 cm), rather than providing each analyte on the corresponding minimum depth resolution for that analyte (e.g. 1 cm for conductivity, 2.5 cm for NH4, 3 cm for H2O2), despite the fact that this underrepresents the resolution of the higher-resolution species like conductivity. Additionally, providing both a conservatively smoothed 3 cm dataset together with the 1 mm resolution dataset allows users to choose the best resolution for their particular uses.
- While the 1 mm resolution dataset is below the "actual" resolution of the dataset when accounting for data smoothing, this high-resolution dataset can be helpful in identifying extreme events in the dataset. Because the 3 cm averaged resolution dataset may smooth extreme events by reducing the amplitude and producing an artificially elevated baseline signal in the vicinity of extremes, it can be useful to provide the higher resolution dataset as a point of reference.

Text has been added to the "Dataset description" section to clarify the true resolution of the record and usefulness of the data at this resolution (1.400-404, 406-409).

On line 31, it is confusing to read a mention of ice cores in the plural when the paragraph begins with the presentation of the new Mount Brown South ice core in the singular (line 29). I think it would be interesting to first present the site and the multiple cores (currently section 1.2) and then introduce Continuous Flow Analysis (currently section 1.1). This would also provide a more logical sequence for the rest of the content. In addition, the sub-section presenting these different cores could be expanded, typically by adding information on the length and position of Alpha, Bravo and Charlie cores in relation to the main core. Consideration should also be given to adding a paragraph (or a dedicated subsection) on what has already been done on these different cores, and possibly summarizing the main findings of Crockart et al. (2021), Jackson et al. (2023) and Vance et al. (2024a). This would also provide more clarity on the addition of this paper compared to the other publications, particularly compared to the Vance et al. paper (2024a) which already mentions the CFA data (even if it's not clear whether Vance et al. use it or not). Finally, section 4.2 on MBS chronology could be added to this part of the text instead of being presented in the data processing section (while it has not been processed in this paper).

Many good points are raised in this comment, which we will address individually point-by-point below:

• It is confusing to read a mention of ice cores in the plural when the paragraph begins with the presentation of the new Mount Brown South ice core in the singular (line 29).

We agree that we worded this in our initial manuscript in a confusing way. We will clarify in the text when we are referring to Mount Brown South (the ice core site), the Mount Brown South ice core array (ice cores plural), and the Mount Brown South ice core (the MBS-Main core specifically).

Edits have been made to clarify as suggested above.

• It would be interesting to first present the site and the multiple cores (currently section 1.2) and then introduce Continuous Flow Analysis (currently section 1.1)... In addition, the subsection presenting these different cores could be expanded, typically by adding information on the length and position of Alpha, Bravo and Charlie cores in relation to the main core.

We will revisit these sections and introduce the ice core site (currently section 1.2) before introducing continuous flow analysis. We will also briefly introduce the Alpha, Bravo, and Charlie shallow cores, referring the reader to the MBS2023 chronology paper (Vance et al., 2024a) for further information, as we do not present any data from these cores in this manuscript.

The sections have been re-arranged, so now the manuscript reads: 1.1 Mount Brown South; 1.1.1 Ice Core Site; 1.1.2 MBS Chronology; 1.1.3 Sample description; 1.2 Continuous flow analysis...

Additional text has been added in section 1.1.1 on the four ice cores (l. 44-46).

• Consideration should also be given to adding a paragraph (or a dedicated sub-section) on what has already been done on these different cores

In lines 31-34 of the introduction, we briefly introduce some of the other works based on the MBS ice cores. As this manuscript is a data descriptor, and discussion of interpretation of other studies performed on the MBS ice cores falls outside the scope of a description of the CFA impurity record. Unless indicated otherwise by the editorial team, we would prefer to refrain from further discussions of other studies in this work.

• ...particularly compared to the Vance et al. paper (2024a) which already mentions the CFA data (even if it's not clear whether Vance et al. use it or not).

The MBS2023 chronology presented by Vance et al. (2024a) was developed prior to and independently of the production of the CFA impurity record presented here, and relied on stable water isotopes and discrete ion chemistry (as stated in l.168-169). We will clarify in these lines that MBS2023 did not use the CFA impurity record, and that this dataset is analytically independent from the data presented in Vance et al. (2024a) and the discrete ion chemistry dataset (Vance et al., 2024b).

Text has been added to the section MBS Chronology including clarification on what datasets were and were not used in its development (lines 67-71)

• Finally, section 4.2 on MBS chronology could be added to this part of the text instead of being presented in the data processing section (while it has not been processed in this paper).

We included section 4.2 in its current place in the text as we believe that the chronology information is most relevant in regards to the usage of the dataset (as we provide a decadally averaged dataset using this chronology), however understand that this could be confusing as the chronology was not produced as part of this manuscript/dataset. We will move the description of the MBS2023 chronology into the introduction, to follow our introduction of the MBS ice core site (what is currently section 1.2).

**The section on the MBS chronology has been moved to section 1.1.2 (page 3).**

Are there any traces of melt layers and wind crusts in the cores? Have their (potential) influences been considered in the various recordings? Even if it has been said that one of the criteria for selecting a site is a minimum of snowmelt in summer, it seems essential to document their occurrence and explain the potential influence of these melt layers and wind crusts on the recordings, especially for the 1 mm resolution dataset.

In the processing of the MBS ice core(s), we have not observed melt layers and in fact have not yet documented any melt layers across the full 295 meter Main core. Certainly, in the upper 20- 25 meters of the ice core, where one would logically expect a potential increase in melt due to a warming climate, we have not recorded any observations of melt, even during summer periods. The annual mean temperature of the MBS site is -27.9 degrees C, and mean summer (DJF) air temperatures are -18.4 - cold enough to preclude melt (Vance et al., 2024a).

Nonetheless, we have observed numerous thin (~Imm thick) ice layers in the core which have been described as "bubble free layers." These features are described in Section 2.6 of Vance et al. (2024a) and are common in Antarctic ice cores (including Law Dome and WAIS among others), and while they have been poorly studied in the past, are an active area of research in the MBS cores. These features have been investigated in the Law Dome ice core in the work of Zhang et al. (2023) and are distinct from melt features. These features are also distinct from "wind crusts," but rather appear to be associated with regional scale atmospheric circulation, such as mid-latitude blocking and meridional moisture transport. Work is ongoing to investigate the chemistry and isotope signatures of these bubble free layers, and their atmospheric drivers. For the purposes of this study, the sub-millimeter scale of these features would be overcome by the internal smoothing of the CFA system, so we do not discuss them in the scope of the CFA dataset.

My major concern is the absence of any discussion of the accuracy and precision of the measurements, which is an essential point in a data paper.

Sub-section 6.3 only presents the limit of detection (LOD), i.e. the smallest concentration that can be measured by the instrument, but not how precise and accurate these measurements are. The relative standard deviation (RSD) - the standard deviation of concentrations of the standards divided by their known levels - of the lowest level standard can be used as a measure of precision, as in Griemann et al. (2022). With regard to accuracy, intermediate standards could be considered as samples (i.e. not used for calibration) and used to define the accuracy of measurements by evaluating the difference between their known concentration and the concentration measured during the run.

We disagree that accuracy and precision are not discussed, perhaps not using this exact wording, but throughout the text (i.e. in sections 5 and 6) we carefully list uncertainties and errors on the data that influence the final accuracy and precision. For example, in Section 4.1 on the depth scale, we discuss the uncertainties on depth scale and provide numbers on these and in section 6 we specifically discuss the analytical precision and data quality. However, in addition to this, we will add the relative standard deviations of the standards used (following Griemann et al. (2022), as suggested) alongside the LOD of the analytes in Table 2, as well as text describing this in Section 6.2.

Standard deviation calculations have been included in what is now Table 2, and descriptive text is included in lines 314-318.

1.267: Have you compared the 3 cm resolution Na+ dataset with Vance et al.'s (2024a) discrete dataset? This could be used as (partial) data quality assessment.

We agree that a comparison to the published discrete dataset would provide an interesting interlaboratory comparison, however we point out that this might be misleading if presented purely as a data quality assessment tool. The discrete measurements were analyzed using ion chromatography, and there is a similar (if not higher) likelihood of e.g. laboratory contamination for these measurements as for the CFA measurements.

We have revised the manuscript to include a comparison with the discrete sodium data from the Vance et al., 2024 dataset, presented in a new section (6.2). We present a visual comparison in an update to Figure 6, as well as a brief discussion and statistical comparison of the two datasets (lines 375-397).

Similarly, no error is mentioned for the third dataset presenting a decadal scale record. A global error calculation - including measurement and dating errors - should be associated with the decadal scale record.

We are unsure what the reviewer means by a "global error calculation" here. However, the errors associated with the measurements are described in the text of the manuscript, which is linked by DOI to the dataset as available from the Australian Antarctic Data Centre. Therefore, users of the decadal scale data set will have easy and open access to the descriptions in this text when accessing and using the dataset. Additionally, further discussion of layer counting error (as well as a comparison to the WAIS and Law Dome ice cores) is thoroughly detailed in the manuscript accompanying the MBS2023 chronology (Vance et al., 2024a), which is referenced in the manuscript text.

Section 4.4 on signal delay time is not detailed enough. Are the numbers and errors the result of statistical analysis? How many estimates have led to these values? How regularly were the bulk and individual delay times measured? And, in practice, how is the "total" delay time applied to the different measurements?

We thank the reviewer for raising the matter of delay time estimation. We will include in the text more detail on the calculation of the bulk delay times based on the individual sample runs, and how many/often standard runs were used to make these calculations.

Additional text has been added in the Signal Delay Time section, including clarification on the bulk delay time (lines 246-247), and a detail of the delay time for each analyte (lines 258-261 and a new table (now Table 1)).

I don't see how section 6.3 (1.256-261) can be used to verify data quality and depth scale accuracy. Firstly, it should be specified what data are involved here (conductivity) and the comparison between the conductivity measured by CFA and the discrete measurements of nssSO4 by Vance et al (2024a) should be clearly mentioned. Secondly, approximating 90 cm as equivalent to 3 years is not correct: it is mentioned in the text that accumulation is highly variable on an interannual scale. Moreover, even if the accumulation were more or less constant, we assume that the authors are using a sliding average of 90 cm on the raw data (observed depth), but density changes with depth, so 90 cm at the surface does not represent the same number of years as at a depth of 200 m when the snow has been compacted into

ice.

Thirdly, I disagree with the statement "we are able to identify many of the volcanoes reported in Vance et al. (2024a)". Looking at Table A2 in the Appendix, only 16 of the 32 volcanoes identified in Vance et al. (2024a) are also present in the conductivity record, i.e. 50%. This deserves an attempt at an explanation (for example, it is possible that the relative proximity of the coast induces a greater presence of impurities resulting in a higher background noise in the conductivity measurements).

Based on these comments, we understand that this section of the text requires some clarifications. We do state in the text of section 6.3 that we are using the CFA conductivity measurements (presented in this manuscript), and comparing to volcanoes identified in Vance et al. (2024a), wherein the methods of volcanic identification are described. We will respond to the individual comments about the content of this section point by point below:

• I don't see how section 6.3 (1.256-261) can be used to verify data quality and depth scale accuracy.

Our intention in this section is to demonstrate the ability of the CFA record described here to correspond with signals identified in the discrete dataset. We consider that a matching eruption signal to within one year depth uncertainty is a basic validation of the general accuracy of the depth scale used to prepare this dataset. We will rephrase the opening of section 6.3 to better describe our intentions with this matching exercise.

Text has been added to section 6.1 Conductivity Peak Matching as described above.

• Firstly, it should be specified what data are involved here (conductivity) and the comparison between the conductivity measured by CFA and the discrete measurements of nssSO4 by Vance et al (2024a) should be clearly mentioned.

It is stated in line 256 that we use conductivity peaks above 3  $\sigma$  in this matching exercise, compared to the data provided in Vance et al. (2024a). We will add text to this section to indicate that the Vance et al., (2024a) dataset uses the non-sea salt sulfate signal to identify volcanic eruption signals.

*Text has been added as described above in section 6.1.*

• Secondly, approximating 90 cm as equivalent to 3 years is not correct: it is mentioned in the text that accumulation is highly variable on an interannual scale. Moreover, even if the accumulation were more or less constant, we assume that the authors are using a sliding average of 90 cm on the raw data (observed depth), but density changes with depth, so 90 cm at the surface does not represent the same number of years as at a depth of 200 m when the snow has been compacted into ice.

We understand that 90 cm does not correspond to exactly 3 years throughout the full core, however, as we are comparing volcanic signal depths identified in the CFA conductivity to the depths of corresponding volcanic signals identified in Vance et al. (2024a), we chose to use a sliding window of fixed depth. We state in the section that this is an "approximation" of 3 years, as this is true as an average approximation across the core. As we have clearly described the accumulation variability characteristic of the site, we consider that describing this as an "approximation" is not inaccurate. We will update the text to clarify that this is indeed only an approximation and does not represent exactly three years throughout the length of the record, due to accumulation variability and density changes downcore.

Text has been added to clarify 90 cm as an approximation and why it was chosen (lines 345-346).

• Thirdly, I disagree with the statement "we are able to identify many of the volcanoes reported in Vance et al. (2024a)". Looking at Table A2 in the Appendix, only 16 of the 32 volcanoes identified in Vance et al. (2024a) are also present in the conductivity record, i.e. 50%.

We will rephrase the text in line 257-258 to the following: "Using this method, we are able to identify 16 of the 32 volcanic events reported in Vance et al. (2024a) to within one year age-at-depth uncertainty."

Proposed change has been made in revised manuscript.

• This deserves an attempt at an explanation (for example, it is possible that the relative proximity of the coast induces a greater presence of impurities resulting in a higher background noise in the conductivity measurements).

The reviewer is correct here, in that as we are comparing bulk conductivity in the CFA record with the discrete non-sea salt sulfate record, there will be inherent differences in the two datasets. However, we have intentionally avoided substantial interpretation of the record here, as ESSD specifies that significant data interpretation is beyond the scope of a data descriptor manuscript.

We have expanded section 6.1 to further discuss the potential causes of the differences in the ability of the two datasets to identify volcanic peaks (lines 352-373), and the possible reasons that the two datasets might return different numbers of identifiable volcanoes at the MBS site. We have further clarified

**Minor comments:**

Some inconsistencies in the text:

Two periods of CFA measurements are mentioned in the text but using different terms (CFA sampling/measurement/melting campaign) (1.54, 1.218-219, 1.238). The same wording should be used to facilitate understanding.

We will update the text to ensure consistent wording throughout.

Proposed change has been made in revised manuscript.

The depths given for dry-drilled and wet-drilled sections are not the same throughout the text: for example, in 1.95-96, dry-drilled and wet-drilled sections are respectively at ~4-94 m and ~95-295 m vs. 5-95 m and 96-295 m in 1.218-219.

We will update the text to ensure that the relevant depth ranges are described accurately and consistently throughout the manuscript.

Proposed change has been made in revised manuscript.

There is a problem with the table numbering. The first Table mentioned in the text is actually Table 2 (line 211). The numbering of the figures and the order in which they are presented should be changed in the text accordingly.

We thank the reviewer for pointing out this error in table numbering. We ensure that the tables are presented in the correct order in the revised manuscript.

Table order has been changed to match the order in which the tables are introduced in the text.

1.2: In the abstract, the authors mention a mean annual accumulation estimated at 20-30 cm ice equivalent. However, 20 cm is never mentioned in the main text. The abstract should only contain information that is present in the main text.

We will update the abstract to reflect only the information in the manuscript.

Proposed change has been made in the revised manuscript.

Figure 1 (near 1.61): some values of lat-lon would be welcome.

We will update Figure 1 to include lat/lon values.

Lat/lon values have been added to the figure.

Legend of Figure 2 (near 1.80): "Shaded areas indicate cuts made with a bandsaw." Do you mean dotted lines?

We thank the reviewer for pointing this out. There is an error in the figure caption for Figure 2. It should state "dotted lines indicate cuts made with a bandsaw. Shaded area (hatched line) indicates surfaces planed for intermediate layer core scanning." We will update the caption to correct this.

*The proposed change has been made in the revised manuscript.*

1.90: A few more details on the scraping of horizontal ends could already be given here: at the very least, mention how much thickness is removed and insist that this is taken into account in the depth logs, especially as one of the datasets is published at millimeter scale.

We will add text at line 91 stating the following: "1-2 mm of ice was removed in this cleaning process. All sample pieces were measured before and after cleaning, and any ice removed in cleaning is accounted for in the depth record."

*The proposed text has been added in the revised manuscript (lines 86-88).*

1.95: The authors should mention why the dry-drilled section only begins at a depth of 4 meters and how the chronology of these first 4 meters is carried out.

As is common with intermediate/deep ice cores, the drilling took place from a trench dug into the firn at the ice core site. This accounts for the 4 meters missing from the top of the Main core

These 4 m were matched with the overlapping shallow cores drilled nearby during the same

season. This is described in the manuscript by Vance et al. (2024a), and as we do not present the data from the shallow cores or the chronology, we refer readers to Vance et al. (2024a) for this information.

1.114: It would be interesting for the readers to mention which type of analysis will be performed in the future.

At time of submission, it is not known exactly which analyses will be performed in the future. As such, we consider this to be beyond the scope of this data descriptor manuscript.

1.137: What are the resolution and accuracy of this cable-driven rotary encoder (in comparison to the previous system)?

The Waycon SX-80 has a sensitivity of  $\gamma = 25$  counts mm-1, and linearity of +/- 0.15%. We will add text that describes the specifications of the instrument at line 137-138.

The proposed change has been made in the revised manuscript (lines 159-160).

l.145: The last sentence of the paragraph raises a question: why would we need more details about the gas extraction system if it was not used for data collection? The answer is only 3 pages later. I suggest adding a few words alluding to the effect of gas extraction on the CFA data presented here.

Thank you for the suggestion. For brevity and to avoid unnecessary repetition, we will amend the last sentence of the paragraph to state "Further detail on the gas extraction system and its impact on measurement quality follows in section 6.1 below."

The proposed change has been made in the revised manuscript.

1.159: What is the impact of inaccuracies of one or two millimeters on the 1 mm dataset? This should be estimated and mentioned in the text.

The impact of inaccuracies on the scale of 1 - 2 mm are described in section 4.1 describing the depth scale. Such inaccuracies would be well below the annual resolution of the record and are unlikely to be of climatic interest to users of this dataset.

l.165: It is mentioned that the procedure for correcting differences in measurements of the same stick is presented in Vance et al. (2024a). However, Vance et al. (2024a) state that "The scaling and shift factors will be described in detail as the CFA trace chemistry and water isotope datasets are developed and published". So it seems that information is missing here.

We apologize for this omission. This was a two-step process: first, the depth scale for the entire Main core needed to be developed taking into account small differences in field to lab core lengths, and then also small differences in the IC (discrete chemistry) stick lengths. This is described in detail in Vance et al. (2024a). However, as the reviewer states, the shift and scaling factors (and CFA datasets) were still under development at the time of publication of Vance et al. (2024a). These discrepancies were accounted for in the assignment of the top depths for each meter-long core section. Further description of the depth scale procedures can be found in Gkinis et al. (2024). We will update the text to more clearly describe the depth adjustment process.

Text has been added to the manuscript (lines 214-221) to more clearly describe this process.

1.184: For readers unfamiliar with CFA measurements, it might be interesting to explain the principle of calibration with standards run at the beginning and end of each measurement. Is it a single calibration with some standards run at the beginning and others at the end? Or two distinct calibrations to account for e.g. measurement drifts?

The standards run procedure is relatively standardized across most CFA systems. In line 188, we refer to Kaufmann et al. (2008), wherein there is a very clear and concise description of the standard calibration procedures, which we use also here. As this is a data descriptor manuscript, and not a methods paper, we will update the text to more specifically refer the reader to the work of Kaufmann et al. (2008).

The proposed change has been made in the revised manuscript (line 232-241).

1.189: Shouldn't the MATLAB script mentioned be made available in a code availability section, as required by the ESSD guidelines?

The MATLAB scripts used here only perform simple calculations in an automated manner. All computations performed in the MATLAB scripts used in data processing are described in the text of the manuscript, and as such, we do not consider the scripts themselves to provide significant additional information to the reader. Additionally, the majority of similar ESSD manuscripts (e.g. Erhardt et al. (2023)) do not include scripts used for simple data processing/calibrations.

*MATLAB* calibration script will be included in manuscript supplement.

1.205-206: Attention should be paid to the significant figures used here (either  $79 \pm 4$  seconds or  $79.x \pm 3.6$  seconds).

We thank the reviewer for pointing this out. We will update the text to ensure consistency of significant digits throughout.

The text has been updated for consistency as described above.

1.254: Is Figure 4 needed? It is not really discussed in the text and is quite redundant with Table 2.

We find Figure 4 to be a helpful means of visualizing the range of variability of the dataset, however we will consider moving it to the supplementary materials if the manuscript length is a concern.

1.279: When mentioning "baseline values", do you mean "true" observed baseline or the limit of detection? It seems more accurate to define the LOD as a threshold.

The baseline values referred to here are the measured MilliQ (Merck RiOsTM16 MilliQ) levels. We will update the text in line 279 to clarify this.

*The proposed change has been made in the revised manuscript (lines 339-340).*

**Figure 5:**

1) The last data point of each complete series reaches 0. If this is an artifact of the plotting (which it probably is), the last point of each series should be removed.

The reviewer is correct that this is a plotting artifact. We will correct this in the revised figure.

Figures have been updated.

2) It looks like there are several points reaching 0 in the H2O2, Na+ and NH4+ series. If this is the case, these points are probably below the LOD/baseline (especially for H2O2 and Na+) and should be removed according to text line 279. It may be interesting to try to plot the LOD for these series (even though they are low for H2O2 and very low for NH4+).

As described in the reply to the comment on Line 279, MilliQ the baseline value used was the threshold for data removal (which will be clarified in the revised text). We will add the LOD to the plot in Figure 5 to provide a visual representation of the information provided in Table 2.

Dashed lines have been added to the plot in figure 5 showing the LOD.

1.286: Is there a reference to confirm that the layer thinning is negligible at this ice core site? This is surprising, given that layer thinning already has a significant effect at 80 m depth in records such as that from Philippe et al. (2016). I also recommend adding the cause of this "layer thinning" (due to strain rates) to the text, to leave no doubt for the reader.

Layer thinning is expected to be small at this site, with a bedrock depth of approximately 2000 metres, as discussed in Vance et al., 2016. However, the reviewer is correct that there will be some layer thinning due to strain across the core, and we will note this in the text. Layer thinning does not affect the use or interpretation of this dataset (which is already on an established chronology), but is of course taken into account when developing accumulation records (regardless of how small layer thinning is expected to be).

*The proposed change has been made in the revised manuscript (line 410).*

**Technical corrections**

Pay attention to sentence syntax (same word used several times in the same sentence):

- 1.2: the second word 'accumulation' can be deleted.
- 1.184: twice the word 'run' in the same sentence.
- 1.223-224: the first part of the second sentence (this gives a smoothing effect) repeats the first sentence (the dynamics lead to smoothing).
- 1.226: 'calculated' used twice in the same sentence.
- 1.227: similar as previous comment with 'using' and 'use'.
- 1.272: similar as previous comment with 'applying' and 'applies'.

We thank the reviewer for pointing these out. We will revise the text to avoid unnecessary repetition.

The proposed changes have been made in the revised manuscript.

Some inconsistencies in the writing:

1.99: lowercase after the ':' (as in 1.7 for example).

There should always be a space between the number and the units (see ESSD guidelines): 1.123, 1.163,

legend of Table 1, 1.227, legends of Figures 4 and 5 (1  $\mu$ m) (e.g. 1.152) The word 'meltrate' is sometimes written 'melt-rate'.

1.178-179: standardize the expression "a # step calibration" or "a # step calibrations".

1.218-219: interval values (in this case, depth values given in brackets) must either be joined to the dash or separated by a space, but must be consistent throughout the text (including in Table 1 or in other intervals like in 1.95-96, 1.139-140, legend of Table 1, 1.227, ...).

1.225: CFA system (with a lower-case s, for consistency).

We thank the reviewer for identifying these typographical errors and inconsistencies. We will update each instance to ensure the text is correct and consistent with ESSD guidelines.

The proposed changes have been made in the revised manuscript.

Some bibliography citations need to be revised:

General question: which order do you use to cite multiple references in text (for example, in 1.112-113)? 1.100: should be "(Bigler et al., 2011)".

1.122: should be "in Dallmayr et al. (2016)."

We will cross check the references to ensure that they are presented correctly and consistently throughout the text.

Purely technical corrections:

1.18: delete the comma before (Legrand and Mayewski, 1997).

We will correct this.

1.122: a point is missing after the citation.

We will correct this.

1.148: "The position derived \*from\* melt-rate data".

This is correct as written, as we refer here to the melt-rate data which was derived from the position of the core at each time step (we will revise this as "position-derived melt-rate" for clarity).

1.199: "(15 seconds for both \*CFA setup\* systems)".

We will revise the text to read "(15 seconds for both the 2018 and 2019 CFA setups)"

1.219: meltrate (instead of metlrate).

We will correct this.

1.220: the second parenthesis ) is missing at the end of the sentence.

We will correct this.

Table 1: min-1 should be in superscript.

We will correct this.

1.229: min-1 should be in superscript.

We will correct this.

1.254: a point is missing at the end of the sentence.

We will correct this.

Legends Figures 4 and 5: perhaps the expression should be "particles > 1 μm per ml".

We will update the figures to use this phrasing.

1.272: I suggest adding a 'by' before 'applying'.

We will correct this.

Table A2: a hyphen is missing at 213.58 m.

We will correct this.

**2 Sample description**

**1.0.1 MBS chronology**

An age scale has been developed for the MBS ice cores, described thoroughly in Vance et al. (2024a). The ice core dating and layer counting was based on a combination of stable water isotopes (Gkinis et al., 2024b) and discrete chemistry measurements. Discrete trace ion measurements were performed using ion chromatography (IC) on ~3 cm samples, using a Thermo-Fisher/Dionex ICS3000 ion chromatograph (for a more detailed description of the discrete impurity measurement methods, see Crockart et al. (2021) and references therein). Chronology development relied primarily the ratio of sulfate to chloride, and there was found to be significant variability in annual layer thickness throughout the length of the core. The chronology was determined by two independent layer counting efforts, one relying on volcanic matching, the other primarily relying on variable chemistry species. These two efforts were followed by careful consideration and joint determination of uncertain years (Vance et al., 2024a). Development of the MBS2023 chronology did not rely on the CFA impurity record, and the CFA dataset presented here is analytically independent from the discrete dataset and chronology presented in Vance et al. (2024b). We encourage direct users of the MBS CFA dataset to use the MBS2023 chronology (Vance et al., 2024b), or any subsequent updates thereafter.

105

**Figure 2.** Schematic diagram showing the cross-section of the MBS main core, with sample sections shown. Shaded areas Dotted lines indicate cuts made with a bandsaw. Dashed lines indicate Shaded area (hatched line) indicates surfaces planed for intermediate layer core scanning. Section shipped to Copenhagen shown outlined in yellow, with CFA stick shaded in blue.

The core was MBS cores were drilled as part of a joint Australian-Danish collaboration. The MBS main core was drilled from a depth of 4.25 m (depth at the top of the borehole, accounting for the recessed floor of the drilling tent and drill trench) to 294.79 m depth using the Danish Hans Tausen ice core drill (9.8 cm core diameter). The ice was transported from the drill site to Hobart, where it was described (in terms of core quality, including breaks, cracks, and other damage to the core), logged, and imaged in the -18°C freezer laboratories at the Institute for Marine and Antarctic Studies. There, it was sectioned lengthwise for analyses. An interior piece with 34 x 34 mm cross section area along the entire length of core was designated for CFA chemistry (Fig. 2). See Vance et al. (2024a) for a full description of the ice processing.

The CFA section, still attached to the wedge-shaped outer edge piece designated for tephra sampling, was transported to Copenhagen, where the samples were prepared for CFA by removing said outer wedge at freezer facilities at the Physics for Ice, Climate and Earth (PICE) at the Niels Bohr Institute (NBI). To minimize potential contamination, the horizontal ends of each sample piece, including any breaks occurring within a sample, were carefully cleaned by scraping with a ceramic blade immediately prior to being placed in frames for melting on the Copenhagen CFA. Approximately 1-2 mm of ice was removed from each break in the ice in this cleaning process. All sample pieces were measured both before and after cleaning, and any ice removed in cleaning has been accounted for in the depth 
[revised manuscript text omitted]

**2.3 Primary differences between 2018 and 2019 CFA setups**

195

205

215

220

As described above, the two system setups used in the MBS CFA measurements are largely similar, using much of the same equipment and instrumentation. The primary differences, and their potential impact on measurements between the two setups are as follows.

- Depth registration system: The 2018 setup utilized a laser distance meter reflecting off of a weight placed on top of the ice, while the 2019 setup used a cable-driven rotary encoder attached to a weight resting on the top of the ice. Additionally, the two instruments result in different amounts of "downtime" in the system during ice stick frame changeover, with slightly more time required when changing frames with the rotary encoder system. While these factors might influence the resulting CFA depth scale, these offsets can be cross-checked with the lengths measured in the freezer during sample preparation. That these differences are minimal, and likely within the scope of the measurement error of the instrumentation.
- Ice stick length: Due differences in depth registration instrumentation (the laser distance meter takes up substantially more vertical height than the draw-wire encoder in the limited freezer height), approximately 55 cm long ice sticks were used in 2018 and 100 cm long sticks were used in 2019. The 55 cm ice sticks introduce additional breaks in the ice which may introduce more potential contamination than with the 100 cm ice sticks. However, the 55 cm ice sticks have the advantage of allowing increased certainty of absolute depths, as the lab-measured top depths are known every 55 cm.
  - Melt rate The ice melt rate was slightly different for the two setups. The impact of the melt rate on sample smoothing
    and resolution is discussed further in section 4.1.
    - Gas extraction system and stable water isotope measurement The 2019 setup introduced both a gas extraction system and an in-line cavity ring-down spectrometer (Picarro L-2140i) for stable water isotope measurements. The gas extraction system is described in more detail in section 5.1, and the stable water isotope measurements are described in Gkinis et al. (2024b). As the gas extraction system required a higher sample volume, the pumped flow rates through the system was different between the two setups (see approximate flow rates in Fig. 3). This resulted in more overflow routed to waste per minute of melting in the 2018 setup than in the 2019 setup. Notably, the target flowrate through each individual measurement instrument is similar for both systems (e.g. ∼1 ml min-1 for each chemistry line and ∼2 ml min-1 for the conductivity and dust measurement lines; target flowrates for each analyte presented in Fig. 3).

**3 Data processing**

**225 3.1 Depth scale**

230

235

240

245

Precise depth information is crucial for ice core data. The position derived melt-rate position-derived melt rate data collected during CFA campaigns is used to reconstruct the total length of ice melted, to which the recorded length of ice removed during preparation must be re-introduced at the appropriate depths. To account for the time when the encoder is removed during the frame changes, for both the laser and draw-wire encoder, the position data is used to identify the times just before and after each frame change period, and the average meltrate melt rate over the previous period of uninterrupted melting is used to reconstruct a continuous melt-rate, melt rate. The down-time that occurs during frame changeover is slightly longer for the draw-wire encoder than that of the laser encoder, as the draw-wire assembly that is attached to the encoder must be fully removed from the remaining ice and subsequently replaced once the new frame has been mounted.

The information recorded during sample preparation is used (including ice removed for decontamination at sample ends and breaks and poor-quality ice not analyzed), together with the position of each break as logged during melting, to create appropriately sized gaps in the depth scale corresponding to the missing ice. The depth scale is finalized by assigning the top of each CFA data run to the recorded top depth of the corresponding bag from the agreed upon accepted field depth measurements. Manual decontamination of the cores can introduce slight (sub-millimeter) discrepancies due to mis-reading lengths on the standard ruler used in preparation as well as slight misjudgments and biases in identification of break positions during melting. These unavoidable inaccuracies are estimated to be on the scale of one or two millimeters at most.

In rare cases, measured lengths of the same stick varied from one measurement to another (for example, when measured in Hobart prior to shipping to Copenhagen). This was found to be due to slightly slanted cuts where one core meets the next, with one measurement taken from the long side and another taken from the short side. These discrepancies were found to amount to less than 0.01 % of the length of the record (Gkinis et al., 2024b). Careful consultation of comprehensive line scan images of the core sections were was used to correct for these errors and determine an agreed upon the most accurate core length across all measurements. This procedure process required the depth scale for the entire Main core to be developed, taking into account small differences in field to lab core lengths as well as any small discrepancies in the IC (discrete chemistry) stick lengths. This is described in detail in Vance et al. (2024a).

**3.2 MBS Chronology**

An age scale has been developed for the MBS ice cores, described thoroughly in Vance et al. (2024a). The ice core dating and layer counting was based on a combination of stable water isotopes (Gkinis et al., 2024a) and discrete chemistry measurements, primarily the ratio of sulfate to chloride, and there was found to be significant variability in annual layer thickness throughout the length of the coreAny such differences were accounted for in the development of the CFA depth scale in the precise assignment of the top depths for each meter-long core section. This top depth assignment was done to prevent any small discrepancies from propagating through the length of the core. Additional description of the depth scale procedures can be found in Gkinis et al. (2024b). The chronology was determined by two independent layer counting efforts, one relying on

volcanic matching, the other primarily relying on variable chemistry species. These two efforts were followed by careful consideration and joint determination of uncertain years (Vance et al., 2024a). We encourage direct users of the MBS CFA dataset to use the MBS2023 chronology (Vance et al., 2024b), or any subsequent updates thereafter.

**260 3.2 Calibrations**

265

285

The absorption and fluorescence spectrometric methods used for chemistry concentration measurements require calibration to convert from instrument signal voltage/light intensity to concentration. In order to properly calibrate these instruments, standard solutions with predetermined chemical concentrations are passed through the system before and after each sample run. A multi-element standard solution is used for a three step calibration of the Ca2+, Na+, and NH4 measurements (Merck Certipur®IC Multi-element Standard VII). A single component standard solutions solution is used for a two step calibrations calibration of H2O2 (Sigma-Aldrich Hydrogen Peroxide Solution 30 wt % in H2O). The multi-element standard solutions are prepared prior to each run, and the peroxide standards are prepared prior to each run from a first dilution prepared at the start of each measurement day. All standards are prepared using ultra-pure deionized water (Merck RiOsTM16 MilliQMilli-Q®). Standard recipes and resulting concentrations are presented in Table A1.

Calibrations are calculated from the standards, which were run at the beginning and end of each measurement run—, as described in Kaufmann et al. (2008) and Erhardt et al. (2023). This allows for monitoring of system drift and provides ample standards data for each run, in case any issues arise during a standards run. Each standard solution is passed through the system in series and the resulting signal voltage was recorded both manually and by the LabVIEW software used to operate the system. For the fluorescence method (Ca2+, NH4+, and H2O2 measurements), the standard calibration is based on the linear relationship between concentration and fluorescence signal voltage. Sodium is a pseudo-absorption method, and calibration is based on a curve fit to the standard signal response (Kaufmann et al., 2008). Calibrations are computed using a semi-automated script written in MATLAB. The script isolates the signal voltage at each standard input concentration and calculates the appropriate calibration coefficients. The standards are well fitted, with the average r-square value for the calibration curves for each analyte being greater than 0.99.

**280 3.3 Signal delay time**

Due to differing distances of each of the measurement instruments from the melter, there is a time delay between when the ice passes the melter and when the measurement takes place.

The time delay between the ice on melter and when the measurement is recorded down stream is calculated in two parts. Firstly, there is the elapsed time from when the sample parcel passes the melter to when it is divided into disparate melt streams for each analytical unit. The bulk melt delay time is measured during sample melting as the elapsed time from when the first sample ice passes the melter (recorded as observed by lab operator) and the initial peak response observed in the conductivity measurements (15 seconds for both systems). Conductivity is used as it is located closest to the melter (by tubing distance/mixing volume as well as time), and thus the first signal to respond. In both systems, the approximate total time offset between the sample ice reaching the melter and the response seen in the conductivity signal was approximately 40 seconds.

Secondly, there is a delay time individual to each species measured due to the internal dynamics of each analytical melt stream. These response delay calculations are computed based on each instrument response time during the standard runruns. This additional delay time is measured as the time elapsed between when the signal increase is seen in the conductivity measurements and target species. We determined that additional delay time as time at which the derivative of the response curve of the standards reaches a maximum (approximating the midpoint of the sample response rise). Delay time varies The average delay time (from all standards runs used in calibration) is presented in Table 1.

Delay times vary for each species: 79, but are similar between the two system setups (Table 1). Other choices for delay calculations could have been to use the start of the rise or even the end of the rise, however due to smoothing can be hard to reliably identify, whereas the maximum of the derivative of the increase is easily and systematically identified. However, we recognize that this can introduce a minor offset between methods that are highly smoothed in comparison to those that have faster response times.

**Table 1.** Average delay time in seconds for each analyte measured as elapsed time between conductivity signal response and analyte signal response, based on the maximum of the first derivative of the signal response measured during each standards run.

|                                  | 2018 (5-95 m depth)                                                    | 2019 (95-295 m depth)                                 |
|----------------------------------|-------------------------------------------------------------------------------|--------------------------------------------------------------|
|                                  | Delay time (seconds)                                                          | Delay time (seconds)                                         |
|                                  | $\underbrace{\text{mean} \pm 3.6 \text{ seconds for std. dev. } (n = 18)}_{}$ | $\underbrace{\text{mean} \pm \text{std. dev. } (n = 29)}_{}$ |
| Ca 2+ <del>, 73</del> | $69 \pm 5.9$ seconds for $9$                                                  | 79 ± 4                                                       |
| $NH_4^+, 48$                     | $66 \pm 5.3$ seconds for $9$                                                  | 74 ± 9
≈ × × × ×                                          |
| $H_2O_2$ , and                   | 62 ± 11                                                                | $68 \pm 4.0 \frac{4}{\text{seconds for 4}}$                  |
| Na + -                | 58 ± 7                                                                 | 48 ± 8                                                |

As the delay times can vary slightly from run to run, due to factors including tubing length, tubing wear, and fluctuations in melt speed, delay times for each species were measured and applied individually to each CFA run of 5-10 m of ice melted. Each delay time estimation was calculated based on the standards data collected either immediately prior to or following the run.

**305 3.4 Calcium data**

310

300

Expected concentrations of calcium at the MBS site are very low (median chemistry concentration from the discrete chemistry measurements is 0.142 ppb). Due to the nature of the system, and the limits of the ultrapure water system used to produce the standards, reagents, and baseline values for measurements produced here, this is significantly below the limit of detection we are reliably able to achieve using the Copenhagen CFA system (see Table 2). As the CFA Ca2+ data for MBS does not show any discernible seasonal cycles throughout the record, and the calibrated Ca2+ concentration values measured close to or below the LOD of our instrumentation, we consider these values too low to report.

**4 Data resolution and smoothing**

**4.0.1 Sampling resolution**

**4.1** Sampling resolution**

Data is registered at one second time steps, and thus the sampling resolution varies with meltrate. The meltrate melt rate of the CFA system is selected to optimize resolution, while accounting for density changes from firn to ice, however with the introduction of the gas extraction system in 2019, the meltrate melt rate was calibrated to produce optimal gas volumes for simultaneous measurement. The 2018 system setup (5 - 95 m depth) CFA setup operated with a target meltrate of 3.5 melt rate of ~4.5 cm min-1 throughout the upper 35 m depth, decreasing across 10 m to reach ~3 cm min-1 below 45 m depth (median actual meltrate melt rate 3.45 cm min-1 across the full 2018 record). The 2019 campaign (96-295 ~95-295 m depth) had a target meltrate melt rate of 4 cm min-1 (median actual meltrate melt rate 4.08 cm min-1. These meltrates). These meltrates produce a direct sampling resolution (independent of smoothing due to response time) of 0.58 and 0.68 mm, respectively.

**4.1.1 Signal dispersion smoothing**

**4.2 Signal dispersion smoothing**

325

330

335

The internal dynamics of the system (mixing volumes and flow conditions within tubing) lead to smoothing due to signal dispersion, as described by Breton et al. (2012). This gives a smoothing effect, wherein a discrete parcel of sample is measured in the CFA System system as a dispersed signal spread across a short time period (on the order of a few seconds) spanning the expected signal response time for that parcel (Breton et al., 2012). This dispersion time is calculated as as an e-folding time calculated based on the 10-9010-90% rise time in the signal response during the standards runs. Using the meltraterun. Based on the melt rate, it is possible to use this dispersion signal time to calculate the effective smoothing length (and thus a minimum realistic resolution) for each species (Table ??). Although the meltrate melt rate used in 2019 was faster than in 2018 (4.08 and 3.45 cm min=1-1 respectively), the shorter response times (likely due to shorter tubing lengths used in the rebuilt system) resulted in a higher resolution in 2019.

It is worth noting that, as described by Breton et al. (2012), the signal peak under realized (dispersed) flow conditions occurs slightly before the expected signal response under idealized "plug flow" conditions. This effect is linked to the specific dynamics of each CFA system, and similar to Erhardt et al. (2022), we do not quantify this slight offset for the Copenhagen system, but note that it could lead to a slight systematic offset (on the order of a few millimeters) biased towards deeper shallower sample depths.

Table 2. Response time as 10%-90% rise time ( $t_{10-90}$ ) Sample data range and e-folding time ( $\tau_e$ ) LOD for all measured species, and resultant effective smoothing length based on meltrate standard deviation (resolutionSD). Values shown of lowest-concentration standard for each of the two system setups from 2018 and 2019 (and the depths measured during each campaign) chemistry analyte.

|                                                | 2018 (5-95 m depth)                 |                                |                             |                               | 2019 (95-295 m                 |       |
|------------------------------------------------|--------------------------------------------|--------------------------------|-----------------------------|-------------------------------|---------------------------------------|-------|
|                                                | Range                                      | Median                         | LOD                  | $\underbrace{SD}$             | Range                                 | Med   |
|                                                | $t_{10-90}$ (s(5th & 95th percentile)      | $	au_e$                        | resolution (cm              | (ppb)                         | $t_{10-90}$ (s(5th & 95th percentile) | Ŧ     |
| Conductivity $(\mu \text{S cm}^{-1})$          | <del>17.5</del> 0.74 - 1.55                | <del>8.0</del> -1.03           | <del>1.01_0.01</del>        | 9.8                           | 4.4 0.57 - 1.38                       | 0.67  |
| $\underbrace{\text{Dust}(\#\text{ml}^{-1})^*}$ | 97.32 - 707.20                             | 281.99                         | 25.00                       |                               | ≂                              | -     |
| Ca 2+ (ppb)*                        | $\bar{\sim}$                               | ₹.                             | 0.82                        | 1.93                          | $\bar{z}$                             | -     |
| NH 4 (ppb)                          | <del>41.7</del> 0.07 - 1.03                | <del>19.0</del> 0.32           | <del>2.39</del> 0.03        | <del>35.3</del> - 1.37 | <del>16.1</del> 0.09 - 0.87    | 2.40  |
| $H_2O_2$ (ppb)                                 | <del>52.9</del> 6.84 - 69.92 | <del>24.1</del> - 26.82 | <del>3.04</del> 0.65 | <del>41.0</del> - 3.31 | <del>18.6</del> -5.40 - 49.01         | 2.792 |
| Na + (ppb)                          | <del>41.6-</del> 2.39 - 38.89              | <del>18.9</del> - 13.54 | <del>2.39</del> 4.26 | <del>33.3</del> - 1.82 | <del>15.1</del> -3.01 - 36.41         | 2.24  |

\* Dashes indicate data not included in the published dataset.

**5 Uncertainties and data qualitylimitations**

**5.1 Gas extraction system interference**

340

345

350

During the 2019 CFA melting campaign, the introduction of a new gas extraction system was trialed. This system setup comprised a gas extraction line originating from the upper outlet of the debubbler, coupled to a micro-module which extracts dry air from the sample stream for gas analysis. These system trials influenced the internal pressure balance of the chemistry system tubing. This had a significant impact on the meltwater acidity (pH) measurements (Kjær et al., 2016), which suffered from back pressure changes influencing the sensitive dye to water ratios, making the pH record unusable. The gas extraction system also impacted the insoluble microparticle data collected by the ABAKUS laser particle counter.

While the specifics of the gas extraction system are beyond the scope of this data description, there is a visible signature in the microparticle record that only occurs when the gas extraction system was on-line with the chemistry CFA. The signature can be seen as a significantly elevated baseline in the microparticle record (measured from Milli-Q@ultrapure water through the system), as well as an alteration in the amplitude of the dust signal. These changes correspond with the recorded valve switches of the gas system. Due to the nature of the system interference, we are not able to reliably correct for this interference. We therefore only present the insoluble microparticle data for the dry drilled section (5-95 m), 5-85 m depth, measured in 2018 before the gas system testing began.

Table 3. Sample data range Response time as 10-90 % rise time  $(t_{10-90})$  and  $\frac{\text{LOD}}{e}$ -folding time  $(\tau_e)$  and resultant effective smoothing length based on melt rate (resolution). Values shown for each analyte of the two system setups from 2018 and 2019 (and the depths measured during each campaign).

|                                                  | 2018 (5-95 m depth)                             |                              |                                       |                         |
|--------------------------------------------------|--------------------------------------------------------|------------------------------|---------------------------------------|-------------------------|
|                                                  | Range (median melt rate = $3.45 \text{ cm min}^{-1}$ ) |                              | Median LOD Range Median LOD           | (median melt rate = 4.0 |
|                                                  | (5th & 95th percentile $t_{10-90}$ (s)                 | $ \underbrace{	au_e}$        | (5th & 95th percentileresolution (cm) | Conductivity (µS cr     |
| Dust (# ml -1 ) * height Conductivity | <del>97.32 - 707.20</del> 17.5                  | <del>281.99</del> 8.0 | <del>25.00</del> _1.01_               | <del>-</del> 9.8        |
| Ca 2+ <del>(ppb)</del> *              | <del>-</del> 46.4                                      | <del>-</del> 21.1            | <del>0.82</del> - 2.71         | <del>-</del> 57.8       |
| NH 4 <del>(ppb)</del>                 | <del>0.07 - 1.03 41.7</del>                            | 0.32-19.0                    | <del>0.03-</del> 2.39                 | 0.09 - 0.87             |
| H 2 O 2 <del>(ppb)</del>   | <del>6.84 - 69.92 5</del> 2.9                   | <del>26.82</del> 24.1 | <del>0.65</del> - 3.04         | <del>5.40 - 49.01</del> |
| Na + <del>(ppb)</del>                 | <del>2.39 - 38.89</del> 41.6                    | <del>13.54</del> -18.9       | 4.26-2.39                             | <del>3.01 - 36.41</del> |

\* Data not included in the published dataset.

**5.2 Analytical precision**

355

360

365

The limit of detection (LOD) varies for each detection channel. LOD Sample concentration ranges, LOD, and SD for each of the analytical detection channels are presented in Table 2. LOD is calculated as 3 times the standard deviation of the blank signal measured on ultrapure (MilliQMilli-Q®) water during the standard runs (Röthlisberger et al., 2000; Gfeller et al., 2014). Sample concentration ranges and limit of detection (LOD) for each of the analytical detection channels is presented in Table 2The standard deviation (SD) of the lowest concentration standard (9.9 ppb for Ca2+ Na+, and NH4 60 ppb for H2O2) is also used as a measure of precision of the system. Frequency histograms for each measured species are presented in Fig. 4.

**5.3 Conductivity peak matching**

To verify data quality and depth scale accuracy, we investigated all conductivity peaks that fall above  $3\sigma$  from a 90-cm (approximately 3 year) moving mean of the conductivity record. Using this method, we are able to identify many of the volcanoes reported in Vance et al. (2024a) to within one year age-at-depth uncertainty. Volcanoes identified in this method include Pinatubo (1991), Agung (1963), Tambora (1815), Mount Mélébingóy/Parker Peak (1640), Ruiz (1595), and Samalas (1257), in addition to a number of the unknown peaks identified in MBS (Table A2). Additional peaks above  $3\sigma$  exist, however have not been matched to volcanoes identified in Vance et al. (2024a).

**5.3 Standards preparation**

370 Measurement uncertainty for the wet chemistry analyses is driven primarily by uncertainty in standards preparation, due to instrument uncertainty of the microliter pipette (20-200 µl Socorex Acura manual®) and bottle-top dispenser (Dispensette®III

**Figure 4.** Relative frequency histograms demonstrating the variability of the impurity data (conductivity ( $\mu$ S cm-1), peroxide (H2O2, ppb), sodium (Na+, ppb), ammonium (NH4+, ppb), and insoluble microparticles (dust, particles per ml >1  $\mu$ m)). Histograms computed based on 3 cm averaged resolution dataset. See Appendix Figure A1 for comparison of CFA and discrete sodium datasets.

5-50 ml, Brand GmbH). This uncertainty is discussed in Gfeller et al. (2014), and as standards used here were prepared in a similar manner, the uncertainty estimate is less than 10 % (Gfeller et al., 2014; Erhardt et al., 2023).

**5.4 Contamination and data cleaning**

Despite careful decontamination of the ice samples prior to melting, some cleaning of the data is still necessary. Often, very short-duration, high concentration spikes can be seen in the data signal due to occasional air bubbles passing through the measurement cells in the instruments despite the use of debubbler and gas permeable membranes (accurelAccurel ®) immediately prior to each detector. These particular signals are easily identified and removed from the record by a simple 10 second smoothing, or (as implemented here) applying by using a filter that applies a threshold cutoff to the differential of the signal (due to the characteristics of these features). The passage of air bubbles through the detectors results in a signal which is characterized

by a significantly steeper increase (and subsequent decrease) in the signal voltage than anything produced by variability in the ice core sample. Because of this characteristic peak shape, we are able to define for the differential of the dataset, a threshold, values above which we use to define "bubble spikes" in the data, which can be safely removed from the dataset.

Contamination signals at core breaks, due to contamination from drill fluid (Estisol-140) or general laboratory contamination, are also relatively easily removed (Erhardt et al., 2022). This type of contamination is characterized by a steep increase in particle count followed by an exponential decay as the contaminated sample passes through the system. For this record, the data coinciding with signals deemed to be caused by this type of contamination have been manually removed from the dataset.

Due to the very low concentrations in Antarctic ice of all species measured and the sensitivity of instrumentation, some measurements fall very close to or below the limit of detection of the system. Data that fall below baseline values (measured from Milli-Q@water run through the system before and after each run) have been removed from the dataset.

**6 Data quality assessments**

385

390

395

400

410

**6.1 Conductivity peak matching**

To verify data quality and depth scale accuracy, and to demonstrate the ability of the CFA record to correlate with signals identified in the discrete dataset, we investigated all conductivity peaks that fall above 3  $\sigma$  from a 90 cm moving mean of the conductivity record. The 90 cm window was chosen as on average it represents an approximation of 3 years, despite the known accumulation variability shown in the MBS cores. Using this method, we are able to identify 16 of the 32 volcanic events reported in Vance et al. (2024a) to within one year age-at-depth uncertainty. Volcanoes identified in this method include Pinatubo (1991), Agung (1963), Tambora (1815), Mount Mélébingóy/Parker Peak (1640), Ruiz (1595), and Samalas (1257), in addition to a number of the unknown peaks identified in MBS (Table A2). This method identifies additional peaks above 3  $\sigma$ , however, these have not been matched to volcanoes identified in Vance et al. (2024a), and may be attributed to factors other than volcanic signals.

We attribute existence of peaks that are not identified in both datasets to the differences in the data types used (comparing conductivity to non-sea salt sulfate) as well as the method of peak identification. Because the peak identification method used here is based solely on bulk conductivity, it is likely that some peaks are related to sources of other soluble ions, in addition to volcanic sulfate. Vance et al. (2024a) use the non-sea salt sulfate, calculated from bulk sulfate following (Plummer et al., 2012), a more specific indicator of volcanic horizons than bulk conductivity, and therefore likely to be better able to distinguish more volcanic horizons. Additionally, Vance et al. (2024a) identified and matched volcanic events qualitatively based on assessment of the size, shape, and relative concentration of sulfate, while we use a simple cutoff threshold method ( $3\sigma$  above the moving mean), which is likely to miss lower-magnitude volcanic horizons that would be more easily identifiable by their shape rather than simply the maximum peak conductivity.

Due to the marginal location of the MBS site, volcanic signals stand out less from the background conductivity record (as a measure of bulk ions) than those of more inland sites (Winstrup et al., 2019). This is due to the proximity to the coastline and increased influence of oceanic sources of impurities leading to dilution of the signal (Plummer et al., 2012; Winstrup et al., 2019; Vance et al., 2019).

Figure 5. Overview of the full CFA record for MBS at 3 cm resolution. Data include electrolytic conductivity of meltwater (Conductivity,  $\mu$ S cm-1), peroxide (H2O2, ppb), sodium (Na+, ppb), (d) ammonium (NH4+, ppb), and (e) insoluble microparticles (dust, particles per ml >1  $\mu$ m). Dashed yellow line indicates LOD for each analyte (see Table 2). Gaps in the dataset indicate sections where ice was removed for decontamination, lost in processing, or where analytical issues resulted in poor data quality.

. When a total ion budget is considered (based on discrete ions), peaks in the total ions seen can be seen that are not associated with elevated sulfate (see Appendix Figure A2).

Jackson et al. (2023) use back-trajectory analysis through the satellite-era to demonstrate that meridional transport pathways are disproportionately represented during the extreme precipitation events that influence MBS accumulation. We hypothesize that this meridional transport pathway might also influence impurity transport to MBS. The greater background influx of impurities due to the coastal location is likely to inhibit the ability of the simple threshold cutoff method used here on the conductivity record to "see" all volcanic horizons able to be identified based on volcanic sulfate.

**Figure 6.** 50 year subset of the 3 cm averaged resolution dataset, covering the period from 1800 to 1850 CE, plotted on the MBS2023 chronology (Vance et al., 2024b). Discrete sodium record shown alongside CFA data for comparison purposes. Data shown correspond with the depth range covering approximately 77.55-63.60 m depth of the data presented in Fig. 5. Note the peaks in both conductivity and insoluble microparticles (dust) corresponding with the influence of the 1815 Tambora eruption, one of the volcanic horizons utilized in the development of the MBS2023 chronology (see Fig. A2).

Accounting for the impact of the different methods used, we consider the ability of the CFA conductivity signal to identify half of the volcanic events found in the non-sea salt sulfate signal by Vance et al. (2024a) across the MBS record to be an external validation of the CFA depth scale and conductivity record described here.

**6.2** Comparison to discrete dataset**

Because discrete measurements have been performed on the full MBS-Main core (Vance et al., 2024b), we are able to use the discrete record as a comparison dataset to investigate similarities and/or differences in the datasets. The only species measured

directly with both methods is sodium, which we use here to cross-check the reliability of our methods and data quality. The published discrete datasets (Vance et al., 2024b) present the raw IC data with minimal data quality assessment beyond the satellite-era section (top  $\sim$ 20 m) of the main core presented in Crockart et al. (2021). As such, there appear to be some outlier samples, likely due to lab contamination, however, a quality assessment of the discrete dataset is beyond the scope of this work. The discrete IC data were measured in micro-equivalents per liter ( $\mu$ eq/L), and have therefore been converted to ppb for comparison to the CFA dataset presented here. In order to produce a direct comparison of the two datasets, the data were resampled (using a simple linear interpolation method) to a common depth scale at 3 cm resolution.

On visual inspection, the two datasets appear to be very well correlated. The seasonal variability of the  $\mathrm{Na^+}$  signal can be clearly identified and peaks can be easily matched between the two datasets. Slight deviations in magnitude can likely be attributed to the differences in measurement method used (ion chromatography vs. fluorescence detection). The CFA method for measuring sodium ions relies on complicated chemical reactions including a reaction column which must be regularly recharged (Kaufmann et al., 2008), and thus minimal drift in the signal is expected. While there are also slight discernible depth discrepancies between the two records, these discrepancies are minor (on the order of a few mm) and do not propagate significantly throughout the record. The discrete  $\mathrm{Na^+}$  dataset is included for visual comparison in Figures 6 as well as Figure A1. To assess variability between the two datasets, we have compared the records using Pearson's correlation, and the two datasets are strongly correlated (r = 0.4307, p > 0.001).

While an inter-lab comparison across these two methods is helpful, it is important to note that issues such as contamination and measurement error are not inherently more likely in one method than the other. Discrepancies between the two datasets, therefore, do not necessarily indicate problems with the CFA dataset presented here. Regardless, the strength of the correlation between the datasets, both in the measured concentration ranges (Fig. A1) and the alignment with depth (Fig. 6), do provide validation and reassurance with regards to the quality of the measurements presented here.

**7 Dataset description**

430

435

440

445

450

455

We present the CFA records in three formats. First, we present the data at 1 mm resolution (full dataset averaged to 1 mm using the CFA depth registration). Although this is in reality an oversampling of the data with regards to the smoothing inherent in the system, we consider this the full dataset at a minimum resolution allowing reduction of instrument noise. While this over-samples the true resolution when accounting for signal smoothing, the 1 mm resolution dataset can be helpful in identifying extreme events in the dataset which may be artificially smoothed by the moving mean method used to produce the 3 cm averaged dataset.

We also present the data on 3 cm resolution (3 cm averaged), which is above the effective resolution of the instrument-smoothed data for the majority of the measured species. Due to the The 3 cm resolution dataset is lower resolution than the "true" smoothed resolution of most of the CFA analytes (all except for peroxide above ~95 m and calcium below ~95 m). However, we chose this resolution so as to provide a cohesive and user-friendly dataset with all species on the same depth scale, despite the fact that this under-represents the resolution of higher-resolution species like conductivity. Additionally, due

[revised manuscript text omitted]

- Vance, T. R., Abram, N. J., Gkinis, V., Harlan, M., Jackson, S., Plummer, C., Segato, D., Spolaor, A., Vallelonga, P., Nation, M. K., Long, C., and Kjær, H. A.: MBS2023 The Mount Brown South ice core chronologies and chemistry data, Ver. 1, Australian Antarctic Data Centre, https://doi.org/doi:10.26179/352b-6298, accessed: 2024-09-07, 2024b.
- Winstrup, M., Vallelonga, P., Kjær, H. A., Fudge, T. J., Lee, J. E., Riis, M. H., Edwards, R., Bertler, N. A. N., Blunier, T., Brook, E. J.,
  Buizert, C., Ciobanu, G., Conway, H., Dahl-Jensen, D., Ellis, A., Emanuelsson, B. D., Hindmarsh, R. C. A., Keller, E. D., Kurbatov, A. V.,
  Mayewski, P. A., Neff, P. D., Pyne, R. L., Simonsen, M. F., Svensson, A., Tuohy, A., Waddington, E. D., and Wheatley, S.: A 2700-year
  annual timescale and accumulation history for an ice core from Roosevelt Island, West Antarctica, Climate of the Past, 15, 751–779,
  https://doi.org/10.5194/cp-15-751-2019, 2019.

---

## Author Response (AR3)

I would like to thank the authors for considering the comments and suggestions made during the first round of reviews. I have a few more minor comments and technical corrections to suggest.

Please note that all line numbers given in the following text refer to the version of the manuscript with the highlighted changes (titled "essd-2024-335-ATC2").

The authors thank the reviewer for taking time to provide additional feedback to improve the manuscript. Please see point-by-point replies below. Line numbering in the replies refers to the revised manuscript submitted (not the tracked-changed version).

**Minor comments:**

Although the authors' decision to omit any discussion of the potential applications of the dataset is clear – invoking that interpretation or analysis of the data presented is not within the scope defined for the ESSD data descriptor manuscript –, it remains regrettable that no mention of these applications has been included. This omission precludes readers from gaining a comprehensive understanding of the extensive potential and richness of such a dataset.

We agree that the dataset is highly useful and understand the importance of highlighting this utility to the scientific community. Text has been added in the introduction to present suggestions for potential use cases for the dataset. Text has been added at lines 31-37 and 39-41.

Regarding the CFA depth scale (1.253-254): "Any such differences were accounted for in the development of the CFA depth scale in the precise assignment of the top depths for each meterlong core section."

Does this mean that there were no differences between the lab and field depths measured for the 55 cm ice sticks?

The wording here is unclear and we thank the reviewer for pointing this out. The full core was cut in the field to approximately one meter long pieces. Hower, due to the CFA freezer configuration during the 2018 campaign, the ~1 m sticks were cut in (approx.) half so that they would fit better in the melter setup. As such, the half-meter sticks were prone to the same field measurement conditions as the meter-long sticks, and the top depth assignment was performed at the top of each meter, despite the added break from dividing each piece. The text has been updated to clarify this in lines 185-191 and 227.

1.293-294: "We determined that additional delay time as time at which the derivative of the response curve of the standards reaches a maximum (approximating the midpoint of the sample response rise)."

Instead, I would suggest using the term "the midpoint of the \*signal\*/or/\*standard\* response rise", as the delay time of each species is defined using the standards signals, not the samples.

We agree with the reviewer here and have updated the text to adopt the following phrasing for enhanced clarity: "approximating the midpoint of the signal response rise"

Legend of Table 2: The description lacks any mention of the median. It is also not mentioned or presented in section 5.2 (1.356-361).

We thank the reviewer for pointing this out. The text has been updated to include reference to the median values in the text and figure captions.

**1.353: The section measured in 2018 goes down to 95 m, not 85 m, right?**

That is correct. However, due to additional concerns with data quality and contamination of the microparticle dataset in the lowermost section of the dry drilled section, we only include depths down to 85 meters. The text has been updated to reflect this (lines 319-321).

1.399: It would be interesting to indicate how many additional peaks this method identifies. This could provide the reader with insight into the (quantitative) differences between the two identification methods. This would be of particular interest in the context of the discussion of bulk conductivity (with the presence of other ion sources), as it would allow for an estimation of the extent to which this phenomenon can influence the results.

This raises an interesting point. The conductivity method described here identified 72 peaks that are more than  $3\sigma$  above a 90 cm moving mean, compared to the 32 volcanic events described in Vance et al. (2024). We appreciate the interest in a means of comparing the qualitative method used by Vance et al. (2024) to the method used in this manuscript. However, the Vance volcanic matching exercise only includes volcanic peaks that were able to be matched with at least one of the other Antarctic records (WAIS divide, Law Dome, and/or Roosevelt Island) based on qualitative matching of the sulfate signal. Thus, there may be volcanic sulfate peaks identified in MBS that were not included in the Vance et al. (2024) volcanic synchronization exercise.

The text has been updated to include the total number of  $>3\sigma$  conductivity peaks identified (72); however, we have refrained from using this to draw any further comparison to the number of volcanic horizons identified in Vance et al. (2024) for the reasons outlined above. See updated text in lines 356-357 and 368-369.

1.442-446: I disagree with using the term "strong correlation" for an r value of 0.43. While it is a significant correlation given the p-value, it is actually rather weak, at best moderate. Empirically, r-values between 0.25 and 0.5 are usually considered to indicate a weak relationship and values between 0.5 and 0.75 a moderate one. Furthermore, regarding the measured concentration ranges, it is clear that the median/average values are substantially higher in the discrete measurements than in the CFA data, even though no numbers are given (only shown in the figure in the appendix). Although I agree that the differences may not necessarily be due to an issue with the CFA data, this should at least be acknowledged in the text.

Thanks to the reviewer for raising this. We have updated the phrasing to indicate that the two datasets are significantly correlated, but that the correlation is moderate to weak. We have also updated the supplementary figure to include the median values of the two datasets and referenced this in the text to add to the comparison between the two datasets.

1.461-462: Please justify the minimal layer thinning at this site by referring to Vance et al. (2016).

We have updated the text to refer to the site description in Vance et al., 2016.

**Technical corrections:**

1.29-30: This sentence is a little confusing, particularly the parenthesis "(together with the Law Dome ice core)", given that MBS is now presented as one of the few ice cores.

The parenthetical has been removed for clarity.

1.207: The "That" in the "That these differences are minimal" could be deleted.

This has been fixed in the text.

1.209: "Due \*to\* differences..."

*This has been fixed in the text.*

1.223: "... target flowrates for each analyte \*are\* presented in Fig. 3."

This has been fixed in the text.

1.272 and others: There are inconsistencies regarding the expression "standards run", which is sometimes written in the plural form and sometimes in the singular (1.272, 1.291, 1.295, Table 1 legend, 1.358).

The text has been updated at each instance to ensure consistency in phrasing.

1.296-298: "Other choices for delay calculations could have been to use the start of the rise or even the end of the rise, however due to smoothing can be hard to reliably identify, whereas the maximum of the derivative of the increase is easily and systematically identified."

Please consider rewriting this sentence. It looks like some words might be missing, and the long sentence makes it harder to understand; in particular, the part "however due to smoothing can be hard to reliably identify".

We agree that this sentence was unclear. It has been updated in the text to read as follows:

"Other choices for delay calculations could have been to use the start or the end of the rise. However, due to smoothing, these points can be hard to reliably identify, whereas the maximum of the derivative of the increase is easily and systematically identified."

1.360: There is a missing comma between Ca2+ and Na+ in the parenthesis.

This has been fixed in the text.

1.414-415: Please rephrase, as the use of the term "seen" twice causes the sentence to sound somewhat peculiar.

This has been fixed in the text.

1.422: There is a missing space between "by" and "Vance".

This has been fixed in the text.

1.440: Change the name of both figures to "Fig." instead of "Figure".

This has been fixed in the text.

1.461: The abbreviation "IE" is not defined in the text. The term "ice equivalent" was used in the abstract and introduction.

*This fixed. The abbreviation IE has been defined at the first instance in the introduction.*